# vPro-MS enables identification of human-pathogenic viruses from patient samples by untargeted proteomics

Marica Grossegesse [1], Fabian Horn [2], Andreas Kurth [3], Peter Lasch [2], Andreas Nitsche[1] & Joerg Doellinger [1,2] ✉

Viral infections are commonly diagnosed by the detection of viral genome fragments or proteins using targeted methods such as PCR and immunoassays. In contrast, metagenomics enables the untargeted identification of viral genomes, expanding its applicability across a broader spectrum. In this study, we introduce proteomics as a complementary approach for the untargeted identification of human-pathogenic viruses from patient samples. The viral proteomics workflow (vPro-MS) is based on an in-silico derived peptide library covering the human virome in UniProtKB (331 viruses, 20,386 genomes, 121,977 peptides). A scoring algorithm (vProID score) is developed to assess the confidence of virus identification from proteomics data (https://github.com/RKI-ZBS/vPro-MS). In combination with diaPASEF-based data acquisition, this workflow enables the analysis of up to 60 samples per day. The specificity is determined to be >99,9% in an analysis of 221 plasma, swab and cell culture samples covering 17 different viruses. The sensitivity of this approach for the detection of SARS-CoV-2 in nasopharyngeal swabs corresponds to a PCR cycle threshold of 27 with comparable quantitative accuracy to metagenomics. vPro-MS enables the integration of untargeted virus identification in large-scale proteomic studies of biofluids such as human plasma to detect previously undiscovered virus infections in patient specimens.

Viruses are ubiquitous in the environment. They are known to infect all domains of life, including bacteria, fungi, plants and animals. Only a small proportion of these viruses can infect humans and cause disease. Increased research activity of viral infections as a result of the SARS-CoV-2 pandemic has once again emphasized that an infection can have long-term consequences, e.g., long COVID, for patients and society beyond the acute infection. At least 10% of people develop long COVID after infection with SARS-CoV-2 including one third without any identified pre-existing medical conditions[1]. The understanding of pathophysiology and risk factors could be supported by more complete surveillance data, such as untargeted proteomics data from SARS-CoV-2 diagnostics. The additional information beyond SARS-CoV-2 identification could be used, e.g., to analyze the immune response, co-infections or early markers in the development of long COVID. However, despite its importance, many viral infections are not diagnosed in detail. One of the consequences is that data on the surveillance of viral infections is incomplete, and the long-term effects of past infections are difficult to recognize. Therefore, there is still a great need to develop diagnostic methods that provide a more comprehensive insight into viral infections in humans.

[1]Robert Koch Institute, Centre for Biological Threats and Special Pathogens: Highly Pathogenic Viruses (ZBS 1), WHO Collaboration Center for Emerging Threats and Special Pathogens, Berlin, Germany. [2]Robert Koch Institute, Centre for Biological Threats and Special Pathogens: Proteomics and Spectroscopy (ZBS 6), Berlin, Germany. [3]Robert Koch Institute, Centre for Biological Threats and Special Pathogens: Biosafety Level-4 Laboratory (ZBS 5), Berlin, Germany. ✉e-mail: Doellingerj@rki.de

In general, methods for virus detection can be divided into two main categories, targeted and untargeted approaches. Targeted methods rely on known genetic markers, known proteins or other characteristics that are unique to a certain taxonomic level. They are suited for situations when an initial hypothesis about the virus causing the patient's symptoms has been formulated or if a certain infection should be excluded. Currently, virus diagnostics is mostly based on targeted approaches, of which real-time quantitative PCR (qPCR) is the gold standard[2,3]. qPCR is characterized by high sensitivity, high specificity and ease of scalability and assay development. On-site diagnostics is dominated by immunoassays, such as lateral flow assays (LFA), which are cheap and rapid but usually less sensitive and potentially less specific than qPCR[4,5]. Both methods offer limited multiplexing capability. In contrast, untargeted approaches attempt to detect species without any prior knowledge or hypothesis regarding their identity. Untargeted identification of viruses can be achieved by metagenomic next-generation sequencing (mNGS), where viral sequences are identified within the total DNA/RNA content of a sample[6,7]. Viral metagenomics has paved the way towards a deeper understanding of the actual diversity of viruses in various environments and is used to diagnose infections with unknown etiology. Although mNGS is gaining popularity, it is still limited by its sensitivity, its cost and the complexity of the laboratory and data analysis workflows[8,9]. In order to overcome these limitations NGS is often coupled with viral enrichment methods, e.g. PCR or hybridization probe capturing[10–13]. Amplicon- and capture-based sequencing lacks the untargeted nature of mNGS, but the multiplexing capacity is higher than that of conventional targeted methods. These methods play an important role in the molecular epidemiology of viral infections as they allow for the direct obtaining of whole virus genomes from patient samples while providing a higher sensitivity than mNGS[8,14].

In microbiology, matrix-assisted laser desorption/ionization (MALDI) time-of-flight (ToF) mass spectrometry (MS) has become the method of choice for rapid, high-throughput and untargeted taxonomic classification of cultured bacteria[15,16]. In virology, no comparable method exists that would allow for fast, simple and inexpensive detection of viruses in an undirected manner. In theory, protein analysis could be an alternative to NGS for untargeted virus detection. However, proteomics has not been widely used for this purpose due to technical limitations, such as low throughput, moderate sensitivity and lack of robustness. With the onset of the SARS-CoV-2 pandemic, an increasing number of proteomic studies were published in which targeted assays were developed, which enabled single virus detection by liquid chromatography and mass spectrometry (LC-MS)[17–19]. Although great technical progress has been made in this direction, the technology is still inferior to current methods for targeted virus detection, especially qPCR, in terms of sensitivity and throughput[20,21]. Instead, we are convinced that the full potential of mass spectrometry for clinical diagnostics of viral infections is unfolded by the untargeted approach. Proteomics has made tremendous progress towards deep and comprehensive analysis of proteins on a large-scale. The combination of data-independent acquisition (DIA), fast gradient liquid chromatography and AI-supported data analysis on the latest generation of MS instruments allows the identification of thousands of proteins even in just 30 s[22]. This technical progress has not been exploited for virus detection yet. Only very few proteomic studies exist that aim to detect viruses in an untargeted or at least multiplexed approach[23]. These studies rely on the use of data-dependent acquisition (DDA) on comparatively slow-scanning Orbitrap instruments, which is currently being rapidly replaced for mere identification and quantification of proteins by DIA performed using fast-scanning time-of-flight (ToF) or Astral mass analyzers[24,25]. The increasing throughput and high sensitivity, as well as the unbiased nature of precursor sampling, make DIA well-suited for virus diagnostics.

In this work, a rapid proteomic workflow was developed, which enables the untargeted detection of human-pathogenic viruses using DIA-MS. It is named vPro-MS (Viral Protein Identification by Mass Spectrometry). All protein sequences of human-pathogenic viruses (>1.4 mio) in the UniProtKB were processed to construct a peptide spectral library (~120,000 peptides, vPro peptide library), which covers the entire human virome and is hence specifically designed for diagnostics. This library and its associated metadata were used to identify viral peptides in DIA-MS data and to subsequently assign the corresponding viruses, including highly pathogenic species like monkeypox virus (MPXV) and Ebola virus (EBOV). The reliability of the virus identification is assessed by a score (vProID score) specifically adapted to untargeted proteomics. This proposed method can serve as a groundwork for virus diagnostics based on the latest technological developments in proteomics. In addition, the data analysis strategy can easily be adopted to detect virus-infected individuals in large-scale proteome cohort studies, thus contributing to the future understanding and monitoring of viral infections in the human population.

## Results

### vPro-MS workflow for rapid and untargeted virus identification

The vPro-MS workflow for untargeted virus detection by proteomics consists of three main parts: sample preparation, LC-MS measurements and data analysis (Fig. 1). The workflow takes about 2 h to perform from sample preparation to result (vPro-MS report) and has a throughput of 60 samples per day (SPD). Samples analyzed in this study were prepared using a slightly modified S-Trap protocol and the resulting peptides were loaded manually onto EvoTips. The 1 h digestion of proteins into peptides is the longest step in the sample preparation procedure and the whole workflow. In general, alternative sample preparation strategies should work similarly well, but care must be taken to select lysis buffers and conditions that effectively inactivate viruses[26]. Samples were analyzed using the 60 SPD method on an Evosep LC system (24 min per sample) and a diaPASEF data acquisition scheme on a timsTOF HT mass spectrometer. This LC-MS system offers the robustness necessary to measure thousands of samples in a routine environment. Peptide sequences were identified from the LC-MS data using DIA-NN and the vPro peptide library of the human virome. Human-pathogenic viruses were detected from the DIA-NN output using the vPro-MS R script and metadata of the vPro peptide library. The reliability of virus detection is monitored by calculating a confidence score (vProID), and the results are summarized in a tabular report. Samples can be analyzed in batch mode, during which the peptide identification with DIA-NN is the speed-limiting step. This step can be computationally parallelized. In our computational setting (Intel® Xeon® Platinum 8160, 24 cores, 192 GB RAM) the data analysis was performed regularly in less than 10 min per sample.

### Construction of the human virome peptide spectral library

The strategy of the vPro-MS data analysis workflow is to construct a peptide spectral library covering the complete human virome (vPro peptide library, Fig. 2). This library is then used to identify peptide sequences in DIA-MS data. The peptide spectral library was constructed from all protein sequences of human-pathogenic viruses available in the UniProt Knowledgebase (1,463,727 proteins, release: 2023_1). Initially, proteins without an associated proteome ID were removed. Structural proteins are best suited for the detection of viruses, as they are the most abundant of all virus proteins[27–29]. Therefore, proteins were filtered according to their GOCC (Gene Ontology Cellular Component) terms and only structural proteins, such as core, envelope and virus membrane proteins, were retained (49,657 proteins). An in silico peptide library was predicted from those proteins using DIA-NN to filter the resulting tryptic peptides according to their detectability in terms of $m/z$ values, retention time and ion mobility (126,788 peptides). The lowest common ancestors (lca) of

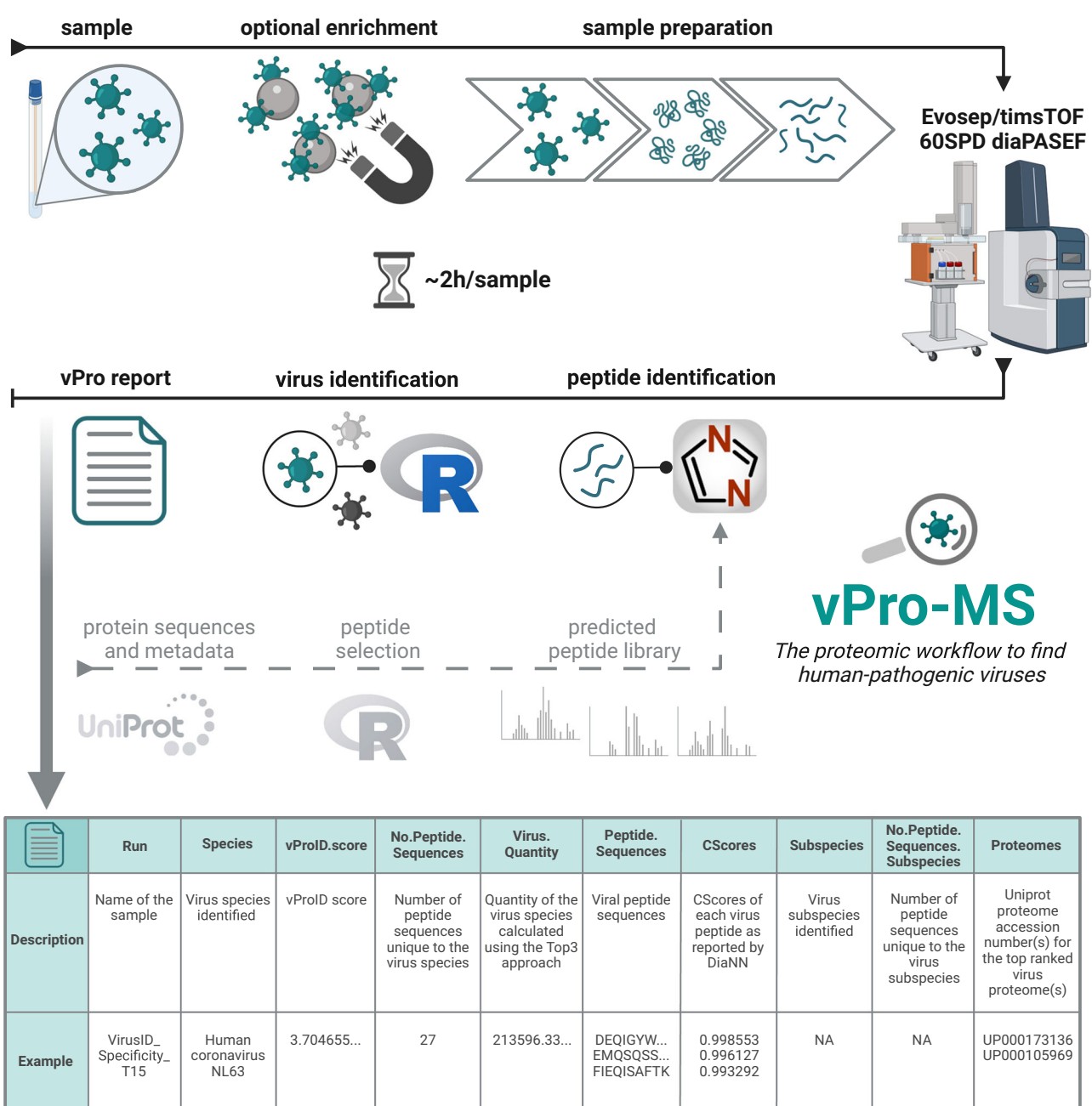

**Fig. 1 | Overview of the vPro-MS workflow for virus identification by untargeted proteomics.** The first step of the vPro-MS workflow is the sample preparation. Proteins are digested into tryptic peptides using S-Trap micro columns and loaded onto Evotips. Afterwards, peptides are analyzed for 24 min per sample, corresponding to a throughput of 60 samples per day (SPD) using diaPASEF on an Evosep One coupled to a timsTOF HT mass spectrometer. Peptide sequences are identified from the MS data using DIA-NN (v 1.8.1) with the vPro peptide library, which is based on UniProt. These peptide sequences are further analyzed by the vPro-MS R script to identify human-pathogenic viruses and generate the vPro-MS report. The confidence of virus identification is assessed by the vProID score. *(Created in BioRender. Doellinger, J. (2025)* https://BioRender.com/8611aej*).*

those peptides were analyzed in R using taxonomic information from UniProt and only species-unique peptides were kept. The International Committee on Taxonomy of Viruses (ICTV) classifies viruses into different hierarchical levels of order, family, subfamily, genus and species. Species may also contain different stable genetic variants called strains or clades. Taxonomic information below the species rank was aggregated into a subspecies rank. This resulted in a species-level annotation for all viruses, which did not necessarily include additional subspecies information. Conversely, all subspecies information is associated with the species information. This is why there are more different subspecies (193) than species (187) in the vPro peptide library. e.g., the species Dengue virus may be connected with the subspecies information Dengue virus type 1, 2, 3, 4 or NA (meaning no subspecies information). Peptides matching to either the human proteome or common contaminants, BSA and trypsin, were removed, and the remaining sequences were annotated with protein-level (proteomeID, UniprotID, gene name, protein name), precursor-level ($m/z$, charge state, iRT, IM, M(ox), CAM) and taxonomic (species, subspecies) information. The peptides were exported as a table (vPro.-Peptide.Library.txt), which served as the database for virus detection in DIA-NN precursor output tables. The library consisted of 121,977 peptide sequences from 331 human-pathogenic viruses (Supplementary Data 1) and was constructed from 20,386 virus proteomes. This represents 94% of all human-pathogenic viruses in the UniProtKB. The

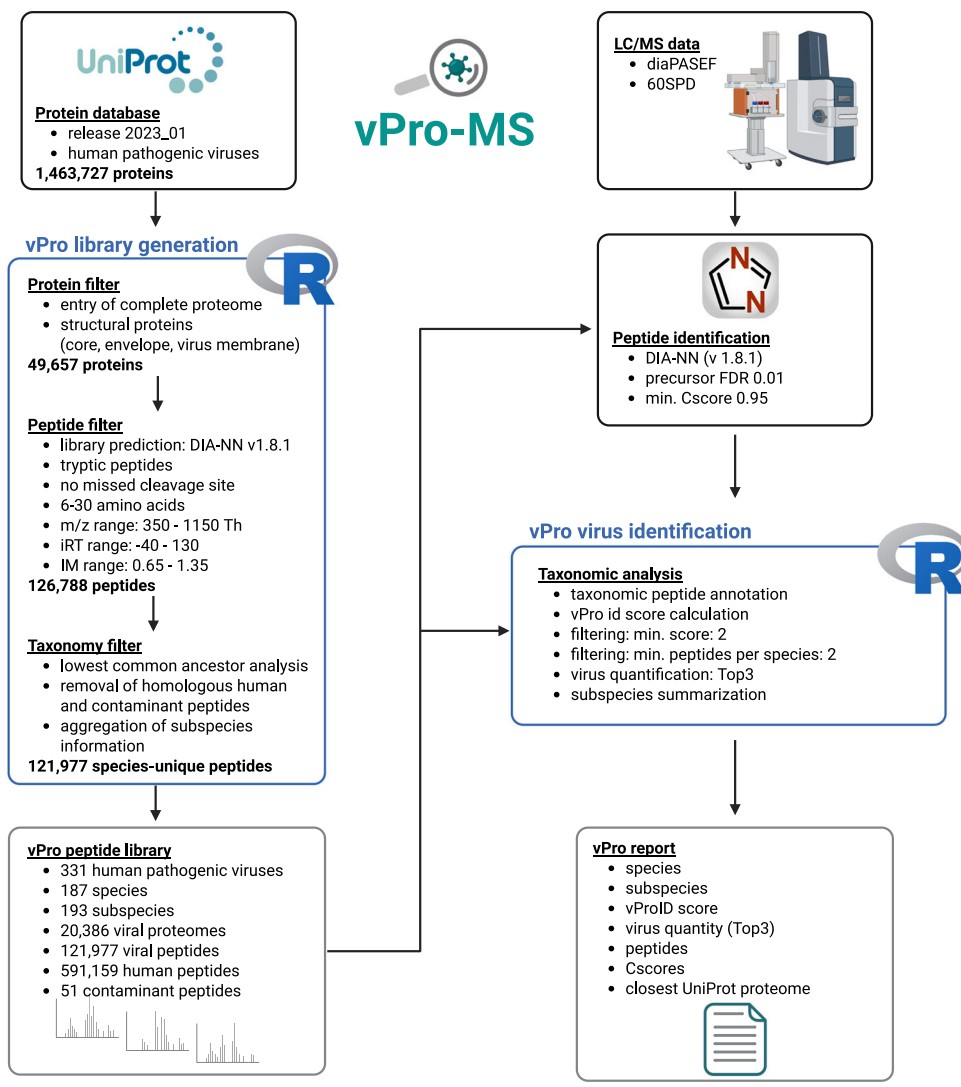

**Fig. 2 | Data flow chart of vPro-MS for virus identification by proteomics.** Data processing of the vPro-MS workflow is split into two parts: peptide library construction and virus detection. At first, a peptide library is generated based on UniProt protein sequences. The UniProt database (release 2023_01) contains >1.4 million protein sequences from human-pathogenic viruses. Structural virus proteins are extracted from these sequences and are used to predict a viral peptide spectral library. Peptides are further filtered for detectability (*m/z*, iRT, IM) and taxonomic specificity. The remaining peptides form the vPro peptide library, to which human and contaminant peptide sequences are added. This library is used to identify peptides from DIA-MS data using DIA-NN. The peptide sequences are analyzed using the vPro-MS R script to identify human-pathogenic viruses. vPro-MS controls the reliability of virus detection by calculating a confidence score (vProID) and summarizes the results in a tabular report. *(Created in BioRender. Doellinger, J. (2025)* https://BioRender.com/j84lltq*)*.

peptides were also exported in fasta format (vPro.Virus.fasta) along with the peptide.fasta files of the human proteome and common contaminants. These files were used to predict a peptide spectral library of the human virome in DIA-NN (vPro-lib.predicted.speclib). A detailed taxonomic summary of the library is provided in Supplementary Data 2.

**Assessing the confidence of virus identification**

The vPro peptide library covering the complete human virome was used to identify peptide sequences in DIA-MS data. This strategy posed a major challenge for post-processing of the identification results in order to achieve a high specificity when trying to identify very few viral peptides in a sample within a library of more than 100,000 viral sequences. Therefore, confidence of virus identification was supervised with the development of the vProID score.

The number of specific peptides varied greatly between the different virus species within the library. HIV-1 was by far the largest part of the library with 68,103 peptides, which equals 56% of all entries. As

peptides are usually identified in proteomics with a controlled error rate (often ~1%), this means that few HIV-1 sequences were identified in almost every sample just by chance. In order to control the reliability of virus identification, we introduced a confidence score, named vProID (Viral Protein Identification). Care was taken to ensure that the sensitivity of virus detection was not impaired. This score is the log10 value of the number of identified peptides for a certain virus proteome divided by the number of expected peptides for this virus proteome just by chance in a specific sample. Afterwards, the top-ranked proteome for each virus species was kept, and a vProID threshold was applied to filter out unreliable virus identifications. Proteomes with a single matching peptide were also removed at this step. An example for filtering the top-ranked virus proteomes per species based on the vProID score is provided in Supplementary Data 3. The discriminatory power of the vProID score to distinguish between random and true virus identifications for the analysis of nasal swab samples is shown in Fig. 3 and Table 1. In total, 58 swab samples positive for SARS-CoV-2 covering a Ct-range of 18−35, and 8 virus-negative swab samples were

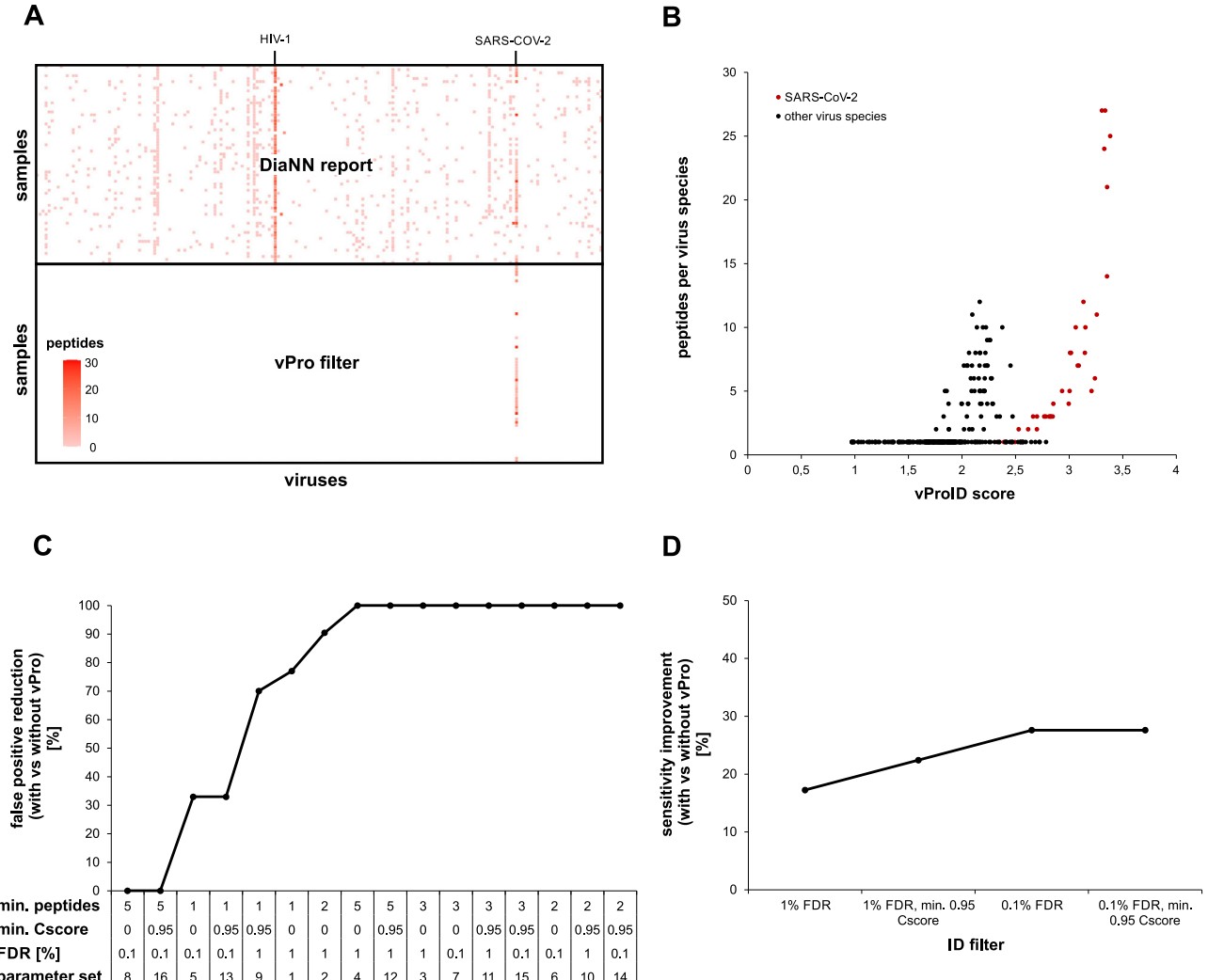

**Fig. 3 | Impact of the vProID score on the sensitivity and specificity of virus identification.** The vPro peptide library covering the human virome (331 viruses) was used to identify human and viral peptides in 66 nasal swab samples, of which 58 were positive for SARS-CoV-2 (ct range 18–35) and 8 were negative. This corresponds to 21,846 individual virus tests within a dataset consisting of 808,704 (redundant) peptide identifications. 16 different parameter sets, including variations of FDR, min. CScore and min. peptides per species, either with or without applying a vProID score threshold, were used to filter the DIA-NN report (see Table 1 for details). Initially, DIA-NN reported the identification of 2361 peptides from 188 different viruses. Applying the vProID score filter on this dataset (parameter set 10), increased the specificity of virus detection to 100 %. The reduction of false positive peptides per virus is visualized in the heatmap (**A**), and the corresponding vPro score distribution is shown as a scatter plot (**B**). The influence of the vProID score on the percentage reduction of false positive virus peptides is shown for all parameter sets in (**C**). The improvement in sensitivity due to the vProID score, with which viruses can be identified with the respective highest specificity, is shown for various parameters in (**D**). For this comparison, the sensitivities of the specified parameters were compared for the minimum number of virus peptides that achieved the highest specificity in each case.

tested for 331 human-pathogenic viruses, which is equivalent to $66 \times 331 = 21{,}846$ individual PCR tests. The DIA-NN report was filtered using 16 different parameter sets including variations of FDR, min. CScore and min. peptides per virus, either with or without applying a vProID score threshold. This resulted in 357 to 2362 viral and between 527,111 and 805,681 human peptide identifications, depending on the filter settings. Please note, that for diagnostic purposes each sample needs to be treated independently and therefore redundant peptide identifications among different samples are summed. The comparison of the DIA-NN report before and after filtering, as well as the vProID score distribution, is visualized in Fig. 3A, B using the example of parameter set 10 (1% FDR, 0.95 min CScore, 2 min. peptides), which was used for the entire remaining manuscript. The specificity on sample- or virus-level and the sensitivity for SARS-CoV-2 detection are reported in Table 1. Specificity is defined as the ability of a test to identify true negatives. As samples were tested for multiple targets

(331 viruses), the specificity of vPro-Ms was calculated by referring to the samples or viruses. Specificity was defined as the number of true negative (TN) virus tests or samples divided by the sum of true negative and false positive (FP) virus tests or samples (specificity = TN/(TN + FP)). The results show that regardless of the parameters, applying the vProID score significantly improved specificity and reduced false positive virus identifications by up to 100% (Fig. 3C). In contrast, when only applying a 1% FDR filter, 959 out of 21,846 individual virus tests were incorrect, which equals about 14 false virus identifications per sample. Without vProID score filtering, perfect specificities of 100% on test and virus-level were only achieved using two parameter sets where a minimum number of 5 peptides per species was required (Table 1). However, the use of such a strict filtering severely reduced the sensitivity by 24%. When the same filter parameters were used in conjunction with vProID scores, perfect specificities on sample and virus-level were achieved requiring only 2 peptides per species. As a

**Table 1 | Impact of the vProID score on the sensitivity and specificity of virus identification for various filtering strategies of peptide identification results**

| Filtering | | | | | Peptide identifications | | | Specificity | | | | | | Sensitivity | | |
|---|---|---|---|---|---|---|---|---|---|---|---|---|---|---|---|---|
| Parameter set | vPro | FDR [%] | Min. CScore | Min. peptides (virus) | Peptides (human) | Peptides (virus) | Peptides (contaminants) | False positive virus tests | Total negative virus tests | Specificity (tests) [%] | False positive samples | Total negative samples | Specificity (samples) [%] | Correct positive samples | Total positive samples | Sensitivity [%] |
| 1 | No | 1 | 0 | 1 | 805,681 | 2361 | 662 | 979 | 21,788 | 95,5 | 8 | 8 | 0 | 37 | 58 | 64 |
| 2 | | | | 2 | 805,681 | 2361 | 662 | 136 | 21,788 | 99,4 | 8 | 8 | 0 | 33 | 58 | 57 |
| 3 | | | | 3 | 805,681 | 2361 | 662 | 73 | 21,788 | 99,7 | 8 | 8 | 0 | 31 | 58 | 53 |
| 4 | | | | 5 | 805,681 | 2361 | 662 | 68 | 21,788 | 99,7 | 8 | 8 | 0 | 21 | 58 | 36 |
| 5 | | 0.1 | | 1 | 527,111 | 357 | 434 | 88 | 21,788 | 99,6 | 4 | 8 | 50 | 34 | 58 | 59 |
| 6 | | | | 2 | 527,111 | 357 | 434 | 32 | 21,788 | 99,9 | 2 | 8 | 75 | 30 | 58 | 52 |
| 7 | | | | 3 | 527,111 | 357 | 434 | 12 | 21,788 | 99,9 | 0 | 8 | 100 | 25 | 58 | 43 |
| 8 | | | | 5 | 527,111 | 357 | 434 | 0 | 21,788 | 100 | 0 | 8 | 100 | 14 | 58 | 24 |
| 9 | | 1 | 0.95 | 1 | 686,735 | 888 | 558 | 314 | 21,788 | 98,6 | 7 | 8 | 13 | 37 | 58 | 64 |
| 10 | | | | 2 | 686,735 | 888 | 558 | 63 | 21,788 | 99,7 | 7 | 8 | 13 | 32 | 58 | 55 |
| 11 | | | | 3 | 686,735 | 888 | 558 | 56 | 21,788 | 99,7 | 5 | 8 | 38 | 29 | 58 | 50 |
| 12 | | | | 5 | 686,735 | 888 | 558 | 40 | 21,788 | 99,8 | 2 | 8 | 75 | 19 | 58 | 33 |
| 13 | | 0.1 | | 1 | 527,111 | 357 | 434 | 88 | 21,788 | 99,6 | 4 | 8 | 50 | 34 | 58 | 59 |
| 14 | | | | 2 | 527,111 | 357 | 434 | 32 | 21,788 | 99,9 | 2 | 8 | 75 | 30 | 58 | 52 |
| 15 | | | | 3 | 527,111 | 357 | 434 | 12 | 21,788 | 99,9 | 0 | 8 | 100 | 25 | 58 | 43 |
| 16 | | | | 5 | 527,111 | 357 | 434 | 0 | 21,788 | 100 | 0 | 8 | 100 | 14 | 58 | 24 |
| 1 | Yes | 1 | 0 | 1 | 805,681 | 2361 | 662 | 225 | 21,788 | 99,0 | 8 | 8 | 0 | 37 | 58 | 64 |
| 2 | | | | 2 | 805,681 | 2361 | 662 | 13 | 21,788 | 99,9 | 3 | 8 | 63 | 33 | 58 | 57 |
| 3 | | | | 3 | 805,681 | 2361 | 662 | 0 | 21,788 | 100 | 0 | 8 | 100 | 31 | 58 | 53 |
| 4 | | | | 5 | 805,681 | 2361 | 662 | 0 | 21,788 | 100 | 0 | 8 | 100 | 21 | 58 | 36 |
| 5 | | 0.1 | | 1 | 527,111 | 357 | 434 | 59 | 21,788 | 99,7 | 2 | 8 | 75 | 34 | 58 | 59 |
| 6 | | | | 2 | 527,111 | 357 | 434 | 0 | 21,788 | 100 | 0 | 8 | 100 | 30 | 58 | 52 |
| 7 | | | | 3 | 527,111 | 357 | 434 | 0 | 21,788 | 100 | 0 | 8 | 100 | 25 | 58 | 43 |
| 8 | | | | 5 | 527,111 | 357 | 434 | 0 | 21,788 | 100 | 0 | 8 | 100 | 14 | 58 | 24 |
| 9 | | 1 | 0.95 | 1 | 686,735 | 888 | 558 | 94 | 21,788 | 99,6 | 3 | 8 | 63 | 37 | 58 | 64 |
| 10 | | | | 2 | 686,735 | 888 | 558 | 0 | 21,788 | 100 | 0 | 8 | 100 | 32 | 58 | 55 |
| 11 | | | | 3 | 686,735 | 888 | 558 | 0 | 21,788 | 100 | 0 | 8 | 100 | 29 | 58 | 50 |
| 12 | | | | 5 | 686,735 | 888 | 558 | 0 | 21,788 | 100 | 0 | 8 | 100 | 19 | 58 | 33 |
| 13 | | 0.1 | | 1 | 527,111 | 357 | 434 | 59 | 21,788 | 99,7 | 2 | 8 | 75 | 34 | 58 | 59 |
| 14 | | | | 2 | 527,111 | 357 | 434 | 0 | 21,788 | 100 | 0 | 8 | 100 | 30 | 58 | 52 |
| 15 | | | | 3 | 527,111 | 357 | 434 | 0 | 21,788 | 100 | 0 | 8 | 100 | 25 | 58 | 43 |
| 16 | | | | 5 | 527,111 | 357 | 434 | 0 | 21,788 | 100 | 0 | 8 | 100 | 14 | 58 | 24 |

**A**

| | specificity panel | SARS-CoV-2 panel | plasma study (PRIDE) | swab study (PRIDE) |
|---|---|---|---|---|
| samples | 27 | 66 | 38 | 90 |
| negative samples (panel) | 8 | 8 | none | 45 |
| negative samples (vPro) | 8 | 8 | n/a | 43 |
| negative virus tests (panel) | 8921 | 21788 | 12540 | 29747 |
| negative virus tests (vPro) | 8918 | 21788 | 12540 | 29745 |
| data source | this study | this study | PRIDE | PRIDE |
| sample types | nasal swabs, cell culture | respiratory swabs | plasma | nasal swabs |
| MS | timsTOF HT | timsTOF HT | Qexactive | timsTOF Pro |
| gradient length | 21 min | 21 min | 45 min | 130 min |
| viruses | CPXV, DENV, HCoV-229E, HCoV-NL63, HCoV-OC43, RSV, IAV, LASV, MARV, MERS, MPXV, RESTV, SARS, SARS-CoV-2, VACV, ZEBOV | SARS-CoV-2 (alpha, delta, omicron) | EBOV | SARS-CoV-2 |

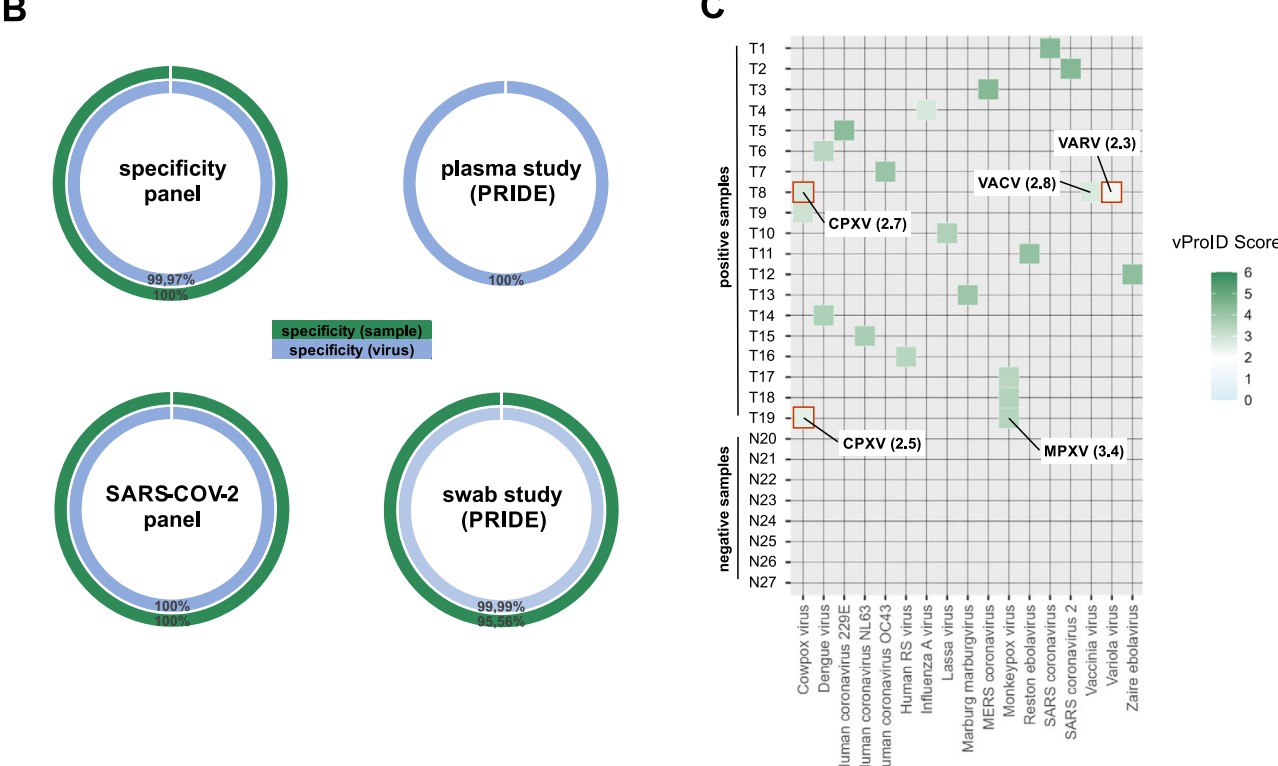

**Fig. 4 | Evaluation of the specificity of vPro-MS for virus identification.** The specificity of the vPro-MS workflow was evaluated by analyzing 221 samples from 4 sources covering 17 different human-pathogenic virus species (**A**). Two sample panels were analyzed by MS for this study and the raw data from two further studies of patient samples were downloaded from PRIDE[30,31]. The sample types included cell-culture supernatants, respiratory swabs and plasma. MS data of all samples were analyzed using the vPro-MS data workflow covering 331 human-pathogenic viruses. This corresponded to the analysis of 73,151 individual virus tests (221 samples tested for 331 viruses). The specificities ranged from 99.97–100% on virus-level and from 95.56–100% on sample-level. The plasma study (PRIDE) does not contain negative samples, and no specificity on sample-level can be calculated (**B**). The results of the specificity panel are further visualized in a heatmap, which displays the vProID score for each species and sample. A minimum vProID score of 2 was required for virus identification (**C**). In two orthopoxvirus samples (T8 and T19), closely related species were identified together with the correct species. The correct species has the highest vProID score in both cases. Wrong virus identifications are outlined in red.

result, vPro improved the sensitivity with which viruses can be identified with maximum specificity up to 28% (Fig. 3D). These results underline, that the vProID score is able to separate random from true virus identifications with high sensitivity even when using a rather moderate FDR of 1% and a highly imbalanced peptide library covering the human virome.

### Evaluation of the specificity of vPro-MS for virus identification
The specificity of the vPro-MS workflow was determined by the analysis of 221 samples from 4 different sources covering 17 human-pathogenic viruses (Fig. 4). Two sample panels were analyzed for this study, and the raw data from two further studies of patient samples were downloaded from the PRIDE repository[30,31]. The sample types included purified viruses, cell-culture supernatants, respiratory swabs and plasma (Fig. 4A). MS data of all samples were analyzed using the vPro-MS data workflow covering 331 human-pathogenic viruses. This corresponded to the analysis of 221 samples x 331 viurses = 73,151 individual virus tests. Specificity was calculated either on virus or sample-level as the number of true negative (TN) results divided by the sum of true negative and false positive (FP) results (specificity = TN/(TN + FP)). The specificities ranged from 99.97–100% on virus-level and from 95.74–100% on sample-level (Fig. 4B). The analysis of the

specificity panel revealed 3 additional virus species identifications in one monkeypox and one vaccinia virus sample (Fig. 4C). These two viruses belong to the same virus family of poxviruses, genus orthopoxviruses. In both cases, the top hit based on the vProID confidence scores was the correct orthopoxvirus species. However, in both samples, either one or two closely related orthopoxvirus species were identified as well. Most probably, this resulted from the incompleteness of the sequence database underlying the vPro peptide library. It might occur that certain tryptic peptides were considered to be species-unique, but in fact, the respective sequence also occurs in closely related species, for which the respective isolates are missing in the UniProt database. Therefore, care should be taken if several species of the same genus are identified in one sample. In the nasal swab study of SARS-CoV-2 patients obtained from PRIDE, SARS-CoV-2 was detected in two samples, which were labeled negative. In both samples, the same two SARS-CoV-2 peptides were identified, which resulted in vProID scores of 2.4 and 2.5, respectively. However, as this study was about analyzing SARS-CoV-2 positive samples, it is unlikely, that these additional identifications were random events. If this were true, it must have been expected that any other species would have been identified. Therefore, it is more likely that those additional identifications were either due to peptide carry-over between samples or resulted from an error in the initial characterization of those samples. Overall, the data underline that the vPro-MS data analysis workflow is highly specific for the detection of human-pathogenic viruses in different sample types and data from various laboratories.

## Evaluation of the sensitivity of vPro-MS for the identification of SARS-CoV-2

The sensitivity of the vPro-MS workflow for the identification of SARS-CoV-2 was evaluated in 66 upper respiratory swab samples covering a Ct range of 18–35 (58 samples) and including 8 virus-negative samples. Additionally, 6 further peptide libraries were created to analyze the influence on the sensitivity (Table 2). In order to analyze the influence of the library size, the initial vPro peptide library (121,977 viral peptides) was either restricted to SARS-CoV-2 (177 viral peptides) or reference proteomes of HIV (53,600 viral peptides). In another library, in conjunction with the SARS-CoV-2 restriction, human entries were restricted to peptides identified in the swab samples, which reduced the number of human entries from 591,159 to 18,655. This library size matches approximately the size of a library obtained from the measurement of synthetic SARS-CoV-2 peptides, which was used to analyze the difference between in silico and measured spectra. Therefore, all theoretical peptides of the SARS-CoV-2 nucleoprotein were synthesized and spiked into negative swab samples to create a SARS-CoV-2 peptide spectral library, including human respiratory swab peptides. In order to further investigate the influence of the library prediction model, we created another library from the same fasta files used for the initial vPro library prediction using the timsTOF model in AlphaPeptDeep instead of the algorithm implemented in DIA-NN[32]. The results demonstrate that the library size as well as the prediction software have only minor effects on the sensitivity for virus identification. However, the library created from measurements of negative samples spiked with synthetic virus peptides clearly outperforms all in silico predicted libraries.

The sensitivity panel contained three different SARS-CoV-2 variants of concern (VOCs), namely alpha, delta and omicron. vPro-MS is currently not able to identify VOCs, because the information needed to include VOCs in the vPro library is not available in UniProt. Nevertheless, in order to analyze the potential of untargeted proteomics for VOC identification, a vPro peptide library with additional VOC entries was created. The vPro reports for all libraries are available in the source data of Table 2. The report of the vPro (SARS-CoV-2 VOCs) peptide library contains additional columns for the taxonomic layer VOC. The VOC was correctly identified only in one delta sample

**Table 2 | Impact of vPro library variations on the sensitivity of SARS-CoV-2 identification**

| Library | | | | | | Summed peptide identifications | | | | Sensitivity | |
|---|---|---|---|---|---|---|---|---|---|---|---|
| # Library | Library | Prediction | Peptide entries (human) | Peptide entries (contaminants) | Peptide entries (virus) | Peptides (human) | Peptides (contaminants) | Peptides (virus) | Peptides reported by vPro (SARS-CoV) | Sensitivity (samples) | Sensitivity (cT) |
| 1 | vPro | DiaNN | 591,159 | 51 | 121,977 | 686,735 | 558 | 888 | 278 | 55 % (32/58) | 27.0 |
| 2 | vPro (SARS-CoV-2 VOCs) | DiaNN | 591,159 | 51 | 122,014 | 687,825 | 566 | 906 | 282 | 55 % (32/58) | 27.0 |
| 3 | vPro (SARS-CoV-2) | DiaNN | 591,159 | 51 | 177 | 697,716 | 573 | 282 | 276 | 52 % (30/58) | 26.1 |
| 4 | vPro (SARS-CoV-2, Human Swab) | DiaNN | 18,655 | 51 | 177 | 743,364 | 744 | 358 | 333 | 52 % (30/58) | 24.8 |
| 5 | vPro (synthetic SARS-CoV-2) | none | 18,655 | 13 | 78 | 796,389 | 655 | 607 | 539 | 74 % (43/58) | 29.4 |
| 6 | vPro (HIV refs) | DiaNN | 591,159 | 51 | 53,600 | 693,068 | 553 | 575 | 282 | 53 % (31/58) | 26.5 |
| 7 | vPro (AlphaPeptDeep) | AlphaPeptDeep | 628,407 | 54 | 117,388 | 771,888 | 591 | 849 | 267 | 52 % (30/58) | 26.4 |

Library descriptions
1 = vPro library
2 = vPro library and entries of SARS-CoV-2 VOCs (alpha, delta, omicron)
3 = library contains only SARS-CoV-2 and human entries
4 = library contains only SARS-CoV-2 and human entries detected in the respiratory swabs
5 = library was built from negative swab samples spiked with synthetic SARS-CoV-2 peptides (no vProID), score and Cscore filter was applied
6 = for HIV only reference proteomes were considered
7 = library was predicted using AlphaPeptDeep (timsTOF model)

**A**     SARS CoV-2 Detection - vPro library

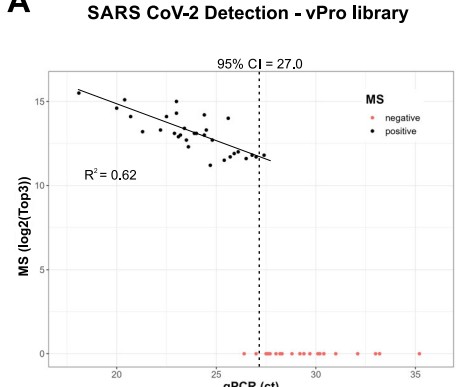

**B**     SARS CoV-2 Detection - synthetic library

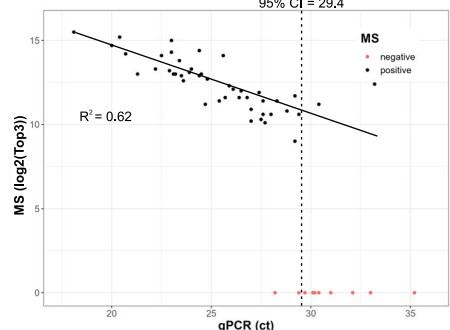

**Fig. 5 | Evaluation of the sensitivity of vPro-MS for the identification of SARS-CoV-2 in nasal swabs.** The sensitivity of vPro-MS for the identification of SARS-CoV-2 was evaluated in 66 respiratory swab samples. The panel included 8 negative samples and 58 SARS-CoV-2 positive samples covering a Ct range of 18–35 of three different variants (alpha, delta, omicron). The quantitative values for proteomics were calculated as the sum of the three most intense peptides (Top3) and compared to qPCR (Ct). Results were compared when using the vPro peptide library (**A**) and a library constructed from the analysis of synthetic SARS-CoV-2 peptides (**B**). The limits of detection are displayed at 95% confidence.

by two unique peptides. If the peptide threshold is reduced to one, already 2 out of all 4 VOC identifications are wrong. The fact that only in 4 out of 58 samples a VOC-unique peptide was identified demonstrates that this approach is not very promising. Maybe more sophisticated approaches might be suited for that purpose, but therefore a well curated protein sequence database of all VOCs is mandatory.

The limit of detection (LOD) with 95% confidence of the initial vPro peptide library corresponded to a Ct value of 27 as determined by qPCR (Fig. 5). It should be noted that approximately twice the amount of starting material (27 µL) was used per qPCR reaction compared to an average volume of 12 µL starting material, which was injected per proteomics measurement. However, twice the sample amount corresponds to a very small Ct difference of just 1 Ct. Virus quantification by proteomics and qPCR correlated with an $R^2$ value of 0.62. This value is well within the range of 0.54 to 0.82, which was reported in recent studies using targeted proteomics to detect SARS-CoV-2[17,18,33–35]. The library constructed from measurements of synthetic SARS-CoV-2 peptides improved the LOD with 95% confidence to a Ct value of 29.4, which is equivalent to ~5 times less virus being detected compared to the in silico predicted vPro peptide library. This underlines the potential of library construction from synthetic peptides to improve the sensitivity in the future. In principle, such specific libraries could also be constructed from measurements of any virus positive sample.

## Discussion

The vPro-MS workflow enables untargeted proteomics to detect human-pathogenic viruses with high specificity in various sample types. Currently, untargeted virus identification is solely based on nucleic acid analysis using mNGS. Protein analysis adds some complementary benefits to the virological toolbox. Proteins are more stable than RNA, which could enable virus detection in samples unsuitable for mNGS. The stability of proteins is the main reason for the increasing interest in proteomics of ancient proteins[36]. The conservation of viral genome sequences can be rather low for certain species, which can complicate the detection of virus variants. This is the reason why amplicon and capture sequencing methods using NGS require frequent adaptations. Protein sequences are more stable than viral genomes as mutations are only partially translated into gene products, which might be beneficial for virus detection in certain cases[37]. Furthermore, mNGS workflows often focus on either RNA or DNA, which are the molecular building blocks of viral genomes. As all

viruses contain proteins, the analysis of DNA and RNA viruses with vPro does not require a customized laboratory and data analysis pipeline. Untargeted proteomics has therefore the capability to enable screening of the human virome with little prior knowledge or customizations.

In addition, untargeted proteomics is less prone to missed virus identifications as a result of emerging sequence variants. This is due to the nature of detection. While qPCR typically detects one or two target regions in the viral genome, untargeted proteomics covers all tryptic peptides detectable by MS. It is also important to note that qPCR only works if the primer and probe binding sequences in the viral genome are intact. This means that they are not entirely mutated or deleted, which reduces the tolerance of qPCR to detect variants. An example is the new MPXV variant clade Ib, which has a deletion in the probe region of the qPCR assay from the CDC, which was used as a reference assay prior to the occurrence of the new variant. Hence, to be able to detect the new variant, a new qPCR assay had to be established[38]. The detection of the new MPXV clade Ib by vPro-MS would not have been affected by the deletion since numerous other peptides would have been available for identification. We evaluated the performance of vPro-MS by collecting data on the sensitivity, specificity and quantitative correlation with qPCR from published mNGS experiments for the detection of SARS-CoV-2 in swab samples (Supplementary data 4)[8,39–42]. The comparison of mNGS and proteomics for the detection of SARS-CoV-2 in swab samples reveals that proteomics is already able to provide a similar specificity on species-level and a comparable quantitative correlation to qPCR as mNGS at similar or even higher throughput. The proteomics workflow is characterized by its comparatively simple sample preparation with low reagent costs and fast and simple data analysis. Contamination is still a major problem in mNGS diagnostics, including cross-contamination (internal contamination) and external contamination[43]. Hence, a defined cut-off to distinguish a positive viral sample from a negative sample with minimal (cross-) contamination reads is mandatory. Recently, tools to identify cross-contamination in mNGS were proposed, but contamination is still one of the reasons why mNGS is not yet widely used for virus diagnostics[44]. The LC-MS system used in this study minimizes the risk of cross-contamination due to the disposable sample spin filters and disposable LC trap columns. However, it is essential to wash the LC columns between samples. We confirmed the absence of run-to-run cross-contamination in the vPro-MS workflow by running alternating measurements of positive and negative SARS-CoV-2 respiratory swab samples (Supplementary data 4). Samples in proteomics are measured and analyzed in a consecutive order. Therefore, the

workflow omits the risk of cross-contamination introduced by multiplexing, which is common in mNGS, and provides high flexibility as sample orders can be prioritized[43]. At the same time, it is uneconomical to sequence a few urgent samples on a high-throughput NGS platform; in proteomics, cost and time scale linearly with sample numbers. An urgent sample can be analyzed in 2 h from swab to report using the vPro-MS workflow whenever needed. The throughput of proteomics is currently limited by the gradient length needed to achieve a certain analysis depth. However, the number of analyzed proteins per time is steadily increasing and has already reached ~1500 unique protein identifications per minute[45]. This continuing progress will increase the throughput of proteomics-based virus diagnostics in the future. A further potential benefit of proteomic-based virus diagnostics compared to mNGS-based analysis is that untargeted proteomics simultaneously provides information about the patient's immune response, which is encoded in the host proteome and not in the host genome.

The current main drawback for virus detection using untargeted proteomics, apart from limited throughput and the instrument's costs leading to high up-front costs, is still the sensitivity. The conclusions drawn from the comparison of sensitivities between different methods applied to different sample cohorts are very limited (Supplementary Data 5). The sensitivities in the mNGS studies were reported as the proportion of samples correctly identified as positive. This does not account for differences in Ct distribution within the sample cohorts. Therefore, the median Ct values should be considered as well. Based on our overview, we hypothesize, that proteomics is in its current stage of development less sensitive than mNGS. However, it should be noted that the sensitivity of mNGS and proteomics depends on the selected throughput. The main focus to develop proteomics into a widely adopted technology for untargeted virus detection must therefore be the improvement of its sensitivity. The sensitivity of vPro-MS will directly benefit from recent technological improvements in proteomics, such as new MS instruments, as the workflow is based on DIA-MS, which has become the most popular and most promising analysis strategy in proteomics[24]. In this study, we demonstrated the possibility of improving the sensitivity of our approach by optimizing the peptide spectral library. We were able to increase the sensitivity to detect SARS-CoV-2 in swab samples 5-fold by constructing a library from synthetic viral peptides. The determined LOD of Ct 29.4 is close to the expected threshold for infectivity of ~Ct 30[46,47]. We expect that the overall sensitivity of our approach will be increased further if we can synthesize all remaining peptides from the human virome. This would require a significant reduction of the human virome library of vPro. This could be achieved by the use of sequence conservation metrics and of artificial intelligence-based prediction of peptide detectability. Another large and untapped potential lies in the adoption of depletion or enrichment strategies during sample preparation, which is common in mNGS[48]. For proteomics, such a strategy has not been developed. This is especially important as many virus samples are stored in virus transport medium (VTM) to preserve virus specimens after collection. This medium contains very large amounts of proteins, namely albumin and collagen, which severely reduce the sensitivity of proteomics. The situation is comparable to plasma proteomics, which has been struggling with a large dynamic range for many years. Recently, several sample preparation protocols were developed, which are able to overcome this limitation[49,50]. Similar strategies could be used to improve the sensitivity of proteomics for untargeted virus detection. Taken together, this study introduces proteomics as an alternative approach for the untargeted identification of human viruses from patient samples. We show that our proposed workflow is flexible, rapid and offers high specificity for virus detection in different sample types, such as plasma and respiratory swabs. vPro-MS has a high potential to improve further its sensitivity, which should enable its wider adoption for high-throughput and rather low-cost screening studies. The data analysis workflow of vPro-MS can further be integrated into large-scale proteome studies of biofluids such as human plasma to explain outliers due to acute infections and to determine the prevalence of persistent infections, such as SARS-CoV-2[50].

## Methods

### Samples

All samples containing viruses were handled in laboratories of biosafety level 2, 3 or 4 in accordance with their risk group classification. For specificity analysis, a panel of 27 cell-culture derived samples were used covering 17 different viruses: Cowpox Virus (CPXV), Dengue Virus serotype 1 (DENV-1), Dengue Virus serotype 2 (DENV-2), Human Coronavirus 229E (HCoV-229E), Human Coronavirus NL-63 (HCoV-NL63), Human Coronavirus OC43 (HCoV-OC43), Influenza A virus (IAV), Lassa virus (LASV), Marburg Virus (MARV), Middle East Respiratory Syndrome Coronavirus (MERS-CoV), Monkeypox Virus (MPXV), Respiratory syncytial virus (RSV), Reston Ebolavirus (RESTV), Severe acute respiratory syndrome coronavirus (SARS-CoV), Severe acute respiratory syndrome coronavirus type 2 (SARS-CoV-2), Vaccinia Virus (VACV), Zaire Ebolavirus (EBOV). The panel further includes 8 negative virus samples. More details of this specificity panel can be assessed in the Supplementary Data 6.

For specificity and sensitivity analysis, a well-characterized panel consisting of 66 upper respiratory swab sample pools was used. This panel contained 50 SARS-CoV-2 PCR-positive samples (alpha variant), 8 SARS-CoV-2 PCR-positive variant samples (omicron and delta variants), as well as 8 SARS-CoV-2 PCR-negative swab samples. The 50 samples were the second version of a panel previously used for validating SARS-CoV-2 antigen rapid tests (panel 1V1 and 1V2) in a larger scale in different laboratories in Germany in combination with a published SARS-CoV-2 variant panel (8 samples) used for the same purpose[4,5,51]. An overview of this sensitivity panel can be found in Supplementary Data 7.

Cell culture-supernatants of LASV, EBOV, RESTV and MARV were diluted 1:4 in inactivation buffer (4% SDS, 35% glycerol, 20% 2-mercaptoethanol and 0.05% bromphenol blue in 200 mM tris, pH 6.8) and heated at 100 °C for 10 min before being removed from the BSL-4 laboratory. All other virus-containing samples were inactivated by addition of 20% SDS solution to a final concentration of 1% and subsequent heating at 95 °C for 10 min. Samples were stored at −20 °C until further use.

### Samples preparation for proteomics

All samples (swab samples, cell-culture supernatants, purified viruses) were provided as liquids. Proteins were processed using S-Trap™ micro columns (PN: C02-micro-80, Protifi, Fairport, NY) according to the manufacturer's instructions with slight modifications (Supplementary Information 1)[52]. Reduction and alkylation of cysteines were not included in this study to reduce time to results, but can generally be integrated as the vPro peptide library contains carbamidomethylated cysteines. It has been shown that reduction and alkylation of cysteines does not affect the depth of proteomic analysis[53]. Proteins were digested for 1 h at 47 °C using 2 μg Trypsin Gold, Mass Spectrometry Grade (PN: V5280, Promega, Fitchburg, WI) per sample in 50 mM TEAB. After elution, peptides were evaporated to dryness, resuspended in 20 μL 0.1% formic acid and quantified by measuring the absorbance at 280 nm using the NanoPhotometer® NP80 (Implen, Munich, Germany).

### Liquid chromatography and mass spectrometry

Peptides were analyzed on an Evosep One liquid chromatography system (Evosep, Odense, Denmark) coupled online via the Captive-Spray source to a timsTOF HT mass spectrometer (Bruker Daltonics, Bremen, Germany). 1 μg of peptides was manually loaded onto Evotips Pure (PN: EV2013, Evosep) and separated using the 60 samples per day (SPD) method on the respective performance column (PN: EV1109,

8 cm × 150 μm, 1.9 μm, Evosep). Column temperature was kept at 40 °C using a column toaster (Bruker Daltonics), and peptides were ionized using electrospray with a CaptiveSpray emitter (PN: 1865710, 20 μm i.d., Bruker Daltonics) at a capillary voltage of 1750 V. Spectra were acquired in diaPASEF mode in the $m/z$ range of 100–1700 and in the ion mobility (IM) range of 0.65–1.35 Vs/cm². Each scan cycle was comprised of 12 diaPASEF scans, which consisted of two IM windows with variable isolation window widths adjusted to the precursor densities using py_diAID covering the $m/z$ range of 350–1150[54]. The collision energy was decreased as a function of the IM from 59 eV at 1/K0 = 1.6 Vs/cm to 20 eV at 1/K0 = 0.6 Vs/cm. The accumulation and ramp times were specified as 100 ms.

## vPro-MS peptide spectral library
Protein sequences of all human-pathogenic viruses (search term: (taxonomy_id:10239) AND (virus_host_id:9606)) were downloaded from UniProt (release 2023_01, 1,463,727 entries)[55]. Gene ontology and taxonomic information of the proteins were also retrieved from Uniprot. At first, the protein sequences were filtered to keep only structural viral proteins according to their gene ontology (GO) annotation (capsid, envelope, virus membrane), which were part of a complete viral proteome using R (version 4.1.3). Afterwards, a peptide library of the remaining protein sequences was predicted using the deep-learning algorithm implemented in DIA-NN (version 1.8.1) with strict trypsin specificity (KR not P) allowing no missed cleavage site in the $m/z$ range of 350–1150 with charges states of 2–4 for all peptides consisting of 7-30 amino acids with enabled N-terminal methionine excision and cysteine carbamidomethylation[56]. The resulting peptides in the spectral library were filtered in R (version 4.1.3) based on retention time (iRT: −40 to 130) and ion mobility (1/$K_0$: 0.65–1.35 Vs/cm²). The remaining peptides were taxonomically annotated based on the organism information of UniProt, which was manually curated to enhance data integrity. Afterwards, the following peptides were removed: (i) peptides, which were not unique to a certain virus species, (ii) peptides mapping to the human proteome, (iii) peptides mapping to common protein contaminants (BSA, trypsin, keratins). The human and contaminant peptide sequences used for filtering are available in the source data folder named vPro-MS Library Generation, along with the manually curated taxonomic information for each UniProt organism entry. Information on the remaining sequences (121,977) was summarized into three separate output files. The Taxonomy.Summary.txt file summarizes the taxonomic information of the library and contains data about the number of proteomes per virus species, as well as the median and maximum number of unique peptides per proteome, and the sum of all unique peptides per species. The vPro.Peptide.Library.txt file contains data for each peptide, including spectral library information (rt, IM, mz, modifications), taxonomic information (species, subspecies) and protein-specific information (gene name, protein Name, UniProtID). It served as the database for virus identification. Furthermore, a vPro.Virus.fasta file was created, in which each peptide sequence has its own entry with a unique identifier. This file was merged with a human (UniProt, UP000005640) and a contaminant peptide fasta file, which was used for predicting the final spectral library by using the deep-learning algorithm implemented in DIA-NN (version 1.8.1)[56]. In order to predict a spectral library from the peptide sequence file, the additional commands --cut and --duplicate-proteins were used.

## Peptide identification
The MS data were analyzed in DIA-NN using default settings including a false discovery rate (FDR) of 1% for precursor identifications with disabled match between run (MBR) option to receive independent outputs for each sample. Mass calibration was disabled and mass tolerance was set to 15 ppm for MS¹ and MS² spectra. The resulting main report was used for virus detection.

## vPro-MS−virus identification
The vPro-MS virus identification script was written in R (version 4.1.3) and can be downloaded from GitHub (https://github.com/RKI-ZBS/vPro-MS). Throughout this manuscript, vPro-MS release 1.0.0 was used. At first, the main report from DIA-NN is loaded, and precursor information is collapsed into peptides after applying a filter (minimum CScore of 0.95). Viral peptides are identified and annotated with taxonomic, proteome- and protein-related information using the vPro.Viral.Peptide.Library.txt file. A virus confidence score (vProID score) is then calculated according to the following equation:

$$\text{vProID score} = \log_{10}\left(\frac{\# \, peptides_{i(sample)} * \# \, peptides_{virus \, and \, human \, (library)}}{\# \, peptides_{sample} * FDR * \# \, peptides_{i(library)}}\right)$$

$i$ = virus proteomes in library

The vProID score is the log10 value of the number of identified unique peptides for a given virus proteome in a certain sample divided by the expected number of unique peptide identifications for this virus proteome just by chance in this specific sample. The vProID score is calculated independently for each virus proteome in the human virome library with a matching peptide sequence in the DIA-NN results. The threshold for virus identification used in this study was set to a minimum of 2 peptides and a vProID score of 2. A vProID score of 2 means that the number of identified peptides for a given viral proteome exceeds the number expected by chance 100-fold. Afterwards, information for all species with at least one proteome above the threshold is aggregated into an output table reporting the viral species and subspecies identified in each sample in conjunction with all matched peptide sequences and the top-ranked proteome. The vPro report represents an independent analysis of each sample from the DIA-NN main report. Viruses are quantified using a Top3 approach[57].

## Synthetic SARS-CoV-2 peptide spectral library
All peptides specific for SARS-CoV-2 in the vPro peptide library ($n = 75$) were synthesized by JPT (JPT Peptide Technologies, Berlin, Germany) either as SpikeTide or MaxiSpikeTide peptides. Afterwards, 100 fmol of each peptide was spiked into 8 different nasopharyngeal swab samples obtained from healthy individuals, which were analyzed by MS using the same settings as described above. The resulting data were analyzed in DIA-NN. At first, a spectral library was predicted from human and SARS-CoV-2 peptides of the vPro-MS library using DIA-NN's deep-learning algorithm with the additional commands --cut and --duplicate-proteins. Precursors were identified from the 8 samples in a combined analysis using the predicted library with default settings and exported as a spectral library, which contained 68 SARS-CoV-2 peptide sequences. This spectral library was used to re-analyze the data from the SARS-CoV-2 panel using default settings in DIA-NN. For SARS-CoV-2 identification, vProID scores were calculated, but no threshold was applied. Instead, a minimum threshold of two unique peptides was required. The library generated from measurement data is very small in contrast to the in silico generated vPro library. It can, therefore, be expected to actually identify a much larger proportion of the library. However, this would mean that the vProID score threshold can be adjusted downwards. However, in order to determine a meaningful threshold for this special case, more data is required than is available in this study.

## Reporting summary
Further information on research design is available in the Nature Portfolio Reporting Summary linked to this article.

## Data availability
The mass spectrometry proteomics data have been deposited to the ProteomeXchange Consortium (http://proteomecentral.proteomexchange.org) via the PRIDE partner repository with

the dataset identifier PXD055135. The source data of tables and figures is published in this study. Source data are provided with this paper.

## Code availability

The vPro-MS code to process data and identify viruses is available at https://github.com/RKI-ZBS/vPro-MS.

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

## Acknowledgements
The authors thank Andreas Puyskens, Caroline Eberle and Juliane Fraissinet for providing the viruses.

## Author contributions
CRediT: Conceptualization: M.G., J.D.; Data curation: J.D.; Formal analysis: J.D.; Funding acquisition, Investigation: M.G., J.D.; Methodology: M.G., J.D.; Project administration: P.L., A.N., J.D.; Resources: A.K., A.N.; Software: F.H., J.D.; Supervision: P.L., A.N., J.D.; Validation: M.G., F.H., J.D.; Visualization: J.D.; Writing—original draft: M.G., J.D.; Writing—review & editing: M.G., F.H., A.K., P.L., A.N., J.D.

## Funding

## Competing interests
The authors declare no competing interests.
