## [Transparent Peer Review file · Nature Communications]

vPro-MS enables identification of human-pathogenic viruses from patient samples by untargeted proteomics

Corresponding Author: Dr Joerg Doellinger

Version 0:

Reviewer comments:

Reviewer #1

(Remarks to the Author)

The authors introduced a new application of DIA proteomics in virus diagnostics by establishing a specific workflow from experiment to analysis all the way to interpretation, named vPro-MS. By the nature of DIA proteomics, it enables the untargeted virus detection with an increased multiplicity. The way they built the comprehensive virus peptide library is novel and provides insights into how mass spec can be used for virus identification. The authors demonstrated the feasibility and specificity of the proposed workflow on several dataset including both home-acquired and publicly available dataset. While even with high specificity at either virus level or sample level, it showed a limited sensitivity compared to targeted qPCR. I agree with the authors that adopting the DIA proteomic method for virus detection is promising and would allow for study of the virus-host interactions, but how to improve the sensitivity of virus detection is the most demanding and challenging work if one wants to put it in real use. Additionally, the non-cheap cost and limited throughput status of DIA data acquisition must be taken into considerations, especially for large-scale diagnostics.

Major points:

1. For the evaluation work, the DIA-NN analysis reports and the processed data which were used in each figure are currently not provided. Please gather these data and make it accessible either as supplementary files or online.
2. In figure3, how was the global peptide identification by sample regardless of the virus identification thresholds? How many peptides came from host, how many from viruses? Can the authors add a supplementary figure to show or describe it somewhere in text? That will provide insights into the identification depth for this specific workflow rather than just virus identification.
3. In figure3, only the peptide count information was displayed to represent the virus identification. vProID score by default is calculated at 1% FDR and requiring at least 2 unique peptides as stated in the method. When assessing how confidently this score can distinguish true viruses from random identifications, it's still necessary to show the vProID score distribution across all viruses. Also regarding the low sensitivity of vPro in SARS-CoV-2 identification (32/58), what does the SARS-CoV-2 vProID scores look like across all samples? For the 16 failed cases, is this because their scores didn't pass the threshold or is there no score available because no peptide were identified?
4. Also in figure3, regarding the sensitivity of SARS-CoV-2 identification, could the authors experiment a bit with the vProID score performance by varying the FDR threshold from 0.1% to 1% and varying the unique peptide count from 2 to 3 or 5.
5. For figure4B, please consider adding a supplement figure to show the uncorrected virus identification along with their corresponding peptide count.
6. For the sensitivity evaluation in figure5, the authors compared the untargeted vPro-MS with the highly sensitive targeted method - qPCR. For a fair comparison, is it possible to include sensitivity comparison results with the untargeted metagenomic method?
7. In the effort to improve the sensitivity of vPro-MS, based on the author's description, the 75 synthetic peptide sequence were obtained from the vPro library and put through a further detectability filtering on samples, resulting in 65 peptides in the final library. (1) the synthetic library is still predicted by DIA-NN like the vPro library, how does the 65 peptides in the synthesized peptide library differ from those in the vPro library in terms of RT, IM and m/z? (2) Even though the synthetic library improved the LOD, it is insufficient to conclude that the improved sensitivity is attributed to synthetic library. Maybe it's due to the reduced library size? The authors could test this by taking the 75 SARS-CoV-2 specific peptides plus the human proteome, creating a predicted library using DIA-NN, and reanalyzing the data to see if it also improves detection sensitivity. (3) Under the hood, DIA-NN performs target extraction of fragment ions based on the library, and the library quality heavily affects its performance, could using better prediction models instead of using DIA-NN improve sensitivity as well? (4) Even

- with the synthetic peptide library, the authors didn't mention whether all positive samples can be identified by vPro-MS.
8. Is vPro-MS able to identify certain type or variants of virus? Taking SARS-CoV-2 panel for example, is it possible to identify the exact variant (alpha, delta or omicron)? The authors need to explicitly discuss this in the manuscript so that the readers will get aware of this limitation or advantage.
 9. As stated in page 15 line 409 'Protein sequences are more stable than viral genomes as mutations are only partially translated into gene products, which might be beneficial for virus detection in certain cases.' When a new variant occurs, immunoassay and qPCR may fail to identify the virus infection. Will vPro-MS have greater tolerance for identifying the virus infection regardless of the exact variant?
 10. Can the authors explain a bit why they decided to use a proteomes-wise level rather than the species or subspecies level when developing the vProID score for virus identification?

Minor points:

1. In page 2 line 45-48, please provide references to the statements 'One of the consequences is that data on the surveillance of viral infections is incomplete and that long-term effects of past infections are difficult to recognize'. As a reader of this manuscript rather than a virus expert, I am curious what kind of clinical cases suffered due to the incomplete surveillance of virus and how it will benefit from a comprehensive diagnostic?
2. In page 2 line 71, please provide reference to 'In microbiology, matrix-assisted laser desorption/ionization (MALDI) time-of-flight (ToF) mass spectrometry (MS) has become the method of choice for rapid, high-throughput and untargeted taxonomic classification of cultured bacteria.' In recent years, many studies were conducted on the Orbitrap instruments. It's hard to conclude which type is the method of choice. If the authors could refer to any supporting references that will help convince readers.
3. In page 3 line 89-91, it would be better to put it more straightforward by rephrasing the sentence in a way.
4. For the virome spectral library construction, initially how many viruses in total were presented in the downloaded Uniport virome proteome database? Please address this in text or label it in figure2.
5. Please keep the library names consistent, 'vPro.Peptide.Library' was used in page 5 line 175, 'vPro.Viral.Peptide.Library' in page 5 line 193, while 'Viral.Peptide.library.txt' in page 8 line 266. Same for 'vPro.Virus.fasta' and 'VirusID.fasta'.
6. In figure4A, please explicitly label how many positive and negative samples in each dataset.
7. In figure4C, peptide number of MPXV is not labeled.
8. In figure5, please color code the samples to distinguish positive from negative.

(Remarks on code availability)

The vPro-MS code is clean and easy to review. The implementation of the proposed score function aligns well with the described approach in the manuscript. A demo data is provided along with the code. Based on the provided readme file, no issues with installing and running the code. No results from the evaluation work are provided by the authors, which limits fully testing and reproducing the results in the manuscript.

Reviewer #2

(Remarks to the Author)

T

The authors described in the manuscript entitled "vPro-MS enables identification of human-pathogenic viruses from patient samples by untargeted proteomics" the development of a strategy to perform untargeted identification of human viruses based on metaproteomics. This is subject of significant relevance and wide clinical applicability, not only on the surveillance of viral infections, but also, combined with classical virology methods such as cell culture and immunofluorescence, in daily routine of clinical laboratories. Its potential advantages and disadvantages compared to metagenomics based on next-generation sequencing are well explored and discussed throughout the text. To my knowledge, the application described in this manuscript is pioneer and its relevance is supported by the analysis of over 200 samples covering 18 different human-pathogenic virus species.

The objectives of the study are clearly identified, the methods are comprehensive in detail, the study presents its results and discussion with clarity and precision, and the conclusions are supported by the data. It is a well written article, but there are some points that should be addressed before the manuscript can be accepted in this peer-review journal.

"Introduction § 3rd: These studies rely on the use of data-dependent acquisition (DDA) on comparatively slow-scanning orbitrap instruments, which is currently being rapidly replaced for mere identification and quantification of proteins by DIA often performed on fast-scanning time-of flight (ToF) mass spectrometers."

This whole sentence should be updated since newer orbitrap-based analyzers rival with ToFs in DIA in terms of speed, and the analyzer recently introduced with Astral is reported to be the faster for DIA analyses.

"Methods - Samples Preparation for Proteomics: Reduction and alkylation of cysteines was not included due to time constraints in this study. "

Explain what were the time constraints.

Comment the possible effects of including reduction and alkylation in the overall data quality and sensitivity of the strategy.

"Methods - vPro-MS Peptide Spectral Library: Afterwards, a peptide library of the remaining protein sequences was predicted using the deep-learning algorithm implemented in DIA-NN (version 1.8.1) with strict trypsin specificity (KR not P) allowing no missed cleavage site in the m/z range of 350 – 1,150 with charges states of 2 – 4 for all peptides consisting of 7-30 amino acids with enabled N-terminal methionine excision and cysteine carbamidomethylation. "

Why is cysteine carbamidomethylation informed to DIA-NN if this fixed modification was not included during sample prep?

“Afterwards, the following peptides were removed: (i) peptides, which were not specific to a certain virus species, (ii) peptides mapping to the human proteome, (iii) peptides mapping to common protein contaminants (BSA, trypsin, keratins).” Provide more details in this section and include the scripts in the supplementary material.

“Results - Assessing the confidence of virus identification: The number of specific peptides varied greatly between the different species within the library. HIV-1 was by far the largest part of the library with 68,103 peptides which equals 56 % of all entries. As peptides are usually identified in proteomics with a controlled error rate (often ~ 1 %), this means that few HIV-1 sequences were identified in almost every sample just by chance.”

Please discuss the reasons why HIV-1 is so overrepresented (e.g. number of proteins, variants, number of deposited sequences associated with clinical routine genotyping).

Could you comment on the effects of a hypothetical HIV data curation/reduction (e.g. selecting the most detectable HIV peptides) to balance the library?

“Evaluation of the sensitivity of vPro-MS for the detection of SARS-CoV-2: Virus quantification by proteomics and qPCR correlated with an R^2 value of 0.62.”

Please compare the correlation (r^2) of vPro-MS and qPCR for the detection of SARS-CoV-2 with previous described targeted assays (which also included correlation with qPCR).

“Discussion: However, it is essential to wash the LC columns between samples. Samples are measured and analyzed in a consecutive order.”

It is crucial to provide experimental evidence (supplementary section) demonstrating that a single blank injection between samples effectively eliminates carry-over effects. To validate this, we recommend reinjecting low Cts SARS-CoV-2 followed by one, two, or three negative controls, and subsequently analyzing the raw data through the established processing workflow.

(Remarks on code availability)

Reviewer #3

(Remarks to the Author)

The manuscript by M. Grossegeisse et al. entitled “vPro-MS enables identification of human-pathogenic viruses from patient samples by untargeted proteomics”, describes an interesting approach to mine DIA-based LMS data to identify viral proteins to aid the diagnosis of viral infections, and to quantify the viral titer (or a proxy thereof). The chosen approach is to analyze purified virus samples, and to curate the known viral protein sequences to come up with a vetted spectral library with peptides from capsid, envelope, virus membrane proteins, supplemented with real data from a subset of viruses.

While the problem addressed in this manuscript is highly relevant and timely, and the chosen approach is interesting and promising, the manuscript is not yet in a shape that would allow for an accept/reject decision.

The following issues (in a somewhat arbitrary order) have to be addressed before decisions can be considered:

- 1) The manuscript is at times hard to follow also due to numerous inconsistencies in the numbers.
 - a. For example, the authors keep stating 331 virus species, but Table S1 lists only 187.
 - b. They list all the viruses they have tested in the material and methods section; unless the authors really want the reader to count the listed viruses, it might be a good idea to list the viruses tested in a supplementary table.
 - c. Statements like “When applying a 1 % FDR on precursor-level, 959 viruses were incorrectly identified, which equals ~ 14 viruses per sample” also don’t help if only 331 viruses are covered. I can make assumptions about the origin of these 959, but the authors should not rely on assumptions from the reader, but should clearly describe what they mean.
 - d. For Figure 4, the authors state “The specificity of the vPro-MS workflow was determined by the analysis of 221 samples from 4 different sources covering 18 human-pathogenic viruses (Fig. 4).” Looking at panel 4c, it was not clear what T1 to T19 stood for. Never mind N20 to N27. Eventually, it dawned on me, that this is the specificity panel, which has 19 positive samples and 8 negative control samples, which was not clearly explained. In short, a much better explanation of the different datasets used for Figure 4 is needed.
 - e. Often, the authors don’t provide all the numbers and/or clear instructions which values were used to calculate certain outcomes. Case in point A) is Table S2: please provide all the values and clear instructions, so that I can recalculate the vProID score. Currently, I have to gather information from various pages. Similarly, Figure 3: plenty of numbers are given, but based on the description, I am not completely clear which numbers made it into the calculation of specificity. Again, I can make assumptions, but that is always a bad starting point, clearly showing that the manuscript is not good enough in its current state.

Speaking of Figure 3: this seems to be a bit of a weak spot. Just comparing the specificity and sensitivity for 0.1% FDR vs. vPro, the values are 99.8% vs. 100% and 34/58 vs. 32/58, i.e., these numbers are so similar that I would not dare to say that vPro is so much better than just tightening the FDR. If these numbers are the best that the authors can muster, I am not sure whether this approach, despite being interesting, warrants publication in Nature Communications.

The authors introduce some concepts, which are not necessarily easily understood for the majority of the proteomics specialists, which handle mammalian samples, which normally assume that one organism has one proteome. This concept

does not apply to viruses and thus need some explanation so that everybody can follow.

Similarly, the 'virus level' and 'sample level' specificity used in Figure 4 is not well described. I thought I had understood it, but the fact that the plasma study (PRIDE) listed 'no data' for the sample level specificity clearly drove home the message that I did NOT understand the difference. Again, better/clearer explanations are clearly needed.

In the vProID Score formula, it is not clear whether "#peptides(sample)" refers to peptides, unique peptides, PSMs or MS2 spectra. Please be more specific.

The vProID score is introduced and explained in great detail (though not very comprehensible at times). However, in Figure 3 and 4, the meat of the paper, the vPro ID score is not used any longer. Some explanation for this decision to omit further use of the vProID score might be useful.

Figure 5: the authors show some decent correlation between protein intensity (log transformed) and Ct value. Assuming that the protein intensities are Log 2 transformed (which is not clearly stated as y-axis label), it is not clear as to why the slope is not 1. Can the authors speculate/discuss this, as the Ct value should also be the equivalent of a log2 transformation, i.e., the linear fit should have a slope of 1?

In summary, this manuscript has potential, but needs major revisions. The current version is hard to follow, and not well structured. The authors might want to rethink which experiments are truly needed to drive their main message home. Similarly, they should enable the readers to follow their arguments and their calculations without the need of gathering information on various pages and/or having to make assumptions.

(Remarks on code availability)

Version 1:

Reviewer comments:

Reviewer #1

(Remarks to the Author)

The authors have addressed all my comments with expanded evaluations and detailed responses. I have a few minor suggestions that may help improve the visualization of their figures, particularly Figure 3, 4 and 5.

In Figure 3, 1) consider making the axes black instead of gray; 2) consider adding parameter combination information at the bottom of the x-axis in Figure 3C, which can be organized into three rows, each representing one parameter type. This change would help clarify what each parameter set represents, making it easier to interpret the results; 3) please consider adding corresponding dots to the line chart so that the position of each data point is clearer.

In Figure 4C, please modify the color scale to make it more distinguishable across different score levels.

In Figure 5, if there is no specific reason for using the x-axis limits of 15 to 40, please consider making plots more compact to reduce empty space.

Other than that, I have no further concerns.

(Remarks on code availability)

Reviewer #2

(Remarks to the Author)

I appreciate the authors' efforts in carefully addressing all the comments and suggestions provided in the first round of review. The authors have provided well-justified responses to each reviewer comment, with appropriate modifications in the text. The revisions have significantly improved the clarity, rigor, and overall quality of the manuscript.

The revised Table 1 notably enhances the manuscript by clearly illustrating the impact of filtering parameters on virus identification, providing much-needed transparency in this analytical approach.

Minor Corrections Suggested:

During my evaluation, I tested the scripts using both the provided demo data and other DIA-NN data freshly generated in my lab and I identified minor inconsistencies that should be addressed in the final version to ensure vPro-MS reproducibility. Line 4 (`use_mit_license()`) had to be commented out for successful installation/execution in R, as it threw an error in a standard R environment.

The default DIA-NN main report was not in a directly usable format for the pipeline. I needed to reformat my lab's DIA-NN-generated TSV files to match the structure of `vPro-MS/demo/data.txt`. Similarly, DIA-NN predicted library outputs required restructuring to align with `demo/library.txt` format.

While the provided Source material (`546235_1_data_set_10646140_smr62m\Source data\vPro-MS Library Generation\Library Calculation.R`) gives some clues about using tidyverse for this manipulation, these preprocessing steps should be integrated into `vPro.R` (as expected based on the manuscript) or provided a standalone preprocessing script.

(Remarks on code availability)

Details are provided in the comments above for the authors

Point-by-point response to the reviewers' comments

Dear Reviewers,

Thank you for the time and effort you have dedicated to reviewing our manuscript. We have carefully revised our manuscript, addressing all comments and suggestions. We have clarified points where there were questions and adjusted accordingly.

We hope our revisions have successfully addressed all the concerns raised during the review process.

Thank you again for your time.

Sincerely,

Dr. Marica Grossegeesse
Dr. Fabian Horn
Dr. Andreas Kurth
Dr. Peter Lasch
Prof. Dr. Andreas Nitsche
Dr. Joerg Doellinger

Responses to Reviewer #1

REVIEWER COMMENTS

Reviewer #1 (Remarks to the Author):

The authors introduced a new application of DIA proteomics in virus diagnostics by establishing a specific workflow from experiment to analysis all the way to interpretation, named vPro-MS. By the nature of DIA proteomics, it enables the untargeted virus detection with an increased multiplicity. The way they built the comprehensive virus peptide library is novel and provides insights into how mass spec can be used for virus identification. The authors demonstrated the feasibility and specificity of the proposed workflow on several dataset including both home-acquired and publicly available dataset. While even with high specificity at either virus level or sample level, it showed a limited sensitivity compared to targeted qPCR. I agree with the authors that adopting the DIA proteomic method for virus detection is promising and would allow for study of the virus-host interactions, but how to improve the sensitivity of virus detection is the most demanding and challenging work if one wants to put it in real use. Additionally, the non-cheap cost and limited throughput status of DIA data acquisition must be taken into considerations, especially for large-scale diagnostics.

Authors' answer: The authors would like to thank the reviewer #1 (R1) for the time and effort in reading the manuscript and are pleased that the potential of the method has been recognised. In particular, we want to thank R1 for the excellent input by the idea of varying the FDR threshold and the other filters, e.g. unique peptide count, which allows the advantages of vPro to be better emphasized. In addition, the influence of different library generation strategies on the sensitivity was analyzed using R1's suggestions. The revised version of the manuscript includes the following major changes, which we believe have significantly improved the manuscript:

- additional data analysis to demonstrate the performance of the vProID score (completely revised fig.3 and table 1)
- revised figure 4 to improve clarity
- advanced sensitivity evaluation using different spectral libraries (shown in table 2)

Major points:

1. For the evaluation work, the DIA-NN analysis reports and the processed data which were used in each figure are currently not provided. Please gather these data and make it accessible either as supplementary files or online.

Authors' answer: We have deposited the MS data, DIA-NN analysis reports and the vPro reports to the ProteomeXchange Consortium (<http://proteomecentral.proteomexchange.org>) via the PRIDE partner repository with the dataset identifier PXD055135 (Username: reviewer_pxd055135@ebi.ac.uk and Password: 3XAHLqKFjzwk). The DIA-NN analysis reports and the vPro reports can be found in the zip folders of the data type "search" separately for each sample panel. Furthermore, we have included source data for the figures and tables in the SI of the revised version of manuscript.

2. In figure3, how was the global peptide identification by sample regardless of the virus identification thresholds? How many peptides came from host, how many from viruses? Can the authors add a supplementary figure to show or describe it somewhere in text? That will provide insights into the identification depth for this specific workflow rather than just virus identification.

Authors' answer: In order to better highlight the performance of the vProID score, we conducted additional data analysis to compare various filtering strategies for the analysis of the SARS-CoV-2 respiratory swab sample panel. In total, 16 different parameter sets were tested either with or without using vProID scores (32 results in total). We present the results in the completely revised figure 3 and the new table 1. In that way it becomes clearer, that regardless of the other parameters, applying the vProID score significantly improves specificity, reduces false positive virus identifications and enhances sensitivity. Please note, that the results differ from the ones presented in the original figure 3. In the first version of the manuscript, the various filter parameters were applied to the data via the vPro-MS analysis script. However, this meant, that the minimum number of peptides per virus was calculated per genome and not per species/subspecies. The revised manuscript now contains results with vPro-MS completely turned off, as one would be able to generate from the DIA-NN output by obvious filtering strategies such as min. number of peptides per virus species. This leads to an increase of unspecific virus identifications when the vProID score is not applied compared to the original manuscript. The virus identification results for all parameter sets can be accessed via the source data folder in the SI. An overview of the peptide identification results separately listed for human, virus and contaminant sequences for each parameter set is given in table 1. This summarizes the data of 2112 (32 parameters x 66 samples) single sample results. The peptide identifications for each sample and parameter set combination can also be investigated in the source data of table 1 within the .txt files, which are named "Results_Summary_PeptideID_...".

3. In figure3, only the peptide count information was displayed to represent the virus identification. vProID score by default is calculated at 1% FDR and requiring at least 2 unique peptides as stated in the method. When assessing how confidently this score can distinguish true viruses from random identifications, it's still necessary to show the vProID score distribution across all viruses. Also, regarding the low sensitivity of vPro in SARS-CoV-2 identification (32/58), what does the SARS-CoV-2 vProID scores look like across all samples? For the 16 failed cases, is this because their scores didn't pass the threshold or is there no score available because no peptide were identified?

Authors' answer: We have completely revised the section describing the vProID score and also added further data analyses. The results are shown in figure 3 and table 1. In figure 3B the score distribution is shown without any additional peptide filtering. The plot shows that true SARS-CoV-2 identifications separate from random hits based on their vProID scores. The figure illustrates the usefulness of the vProID score for virus identification. Table 1 shows, that at least 1 SARS-CoV-2 peptide was identified in 37 of the 58 positive samples by DIA-NN if no further filter was applied, while not a single SARS-CoV-2 peptide could be identified in 21 of the samples. Please note, however, that the ratio of correctly identified samples (e.g. 32/58) is not a measure of the sensitivity of vPro-MS. We could have simply selected 58 samples with a lower cT value and then achieved 100% identification. Nevertheless, the workflow would not be any more sensitive. The samples were selected to cover a large cT range so that an exact calculation of the sensitivity relative to the gold standard of qPCR is possible. The data in table

I also illustrate that, in principle, error-free identification based on just one peptide is not possible. The table shows that even with an FDR of 0.1% and a min. CScore of 0.95, a specificity of 100% is still not achieved, a limitation that is inherent in the current FDR-based approach to peptide identification. It is not possible to perform further calculations with only one sequence. If two SARS-CoV-2 peptides were identified by DIA-NN, the sample passed the threshold for virus identification by vPro-MS.

4. Also in figure3, regarding the sensitivity of SARS-CoV-2 identification, could the authors experiment a bit with the vProID score performance by varying the FDR threshold from 0.1% to 1% and varying the unique peptide count from 2 to 3 or 5.

Authors' answer: Excellent suggestion, see answer on your major point 2. We included the respective analysis and show the results in the revised figure 3 and table 1.

5. For figure4B, please consider adding a supplement figure to show the uncorrected virus identification along with their corresponding peptide count.

Authors' answer: We have added the vPro reports of all sample panels in the SI. The peptide counts of the incorrect virus identifications in the SARS-CoV-2 study (PRIDE) are now also mentioned in a new sentence on page 14: "In both samples the same two SARS-CoV-2 peptides were identified, which resulted in vProID scores of 2.4 and 2.5 respectively."

6. For the sensitivity evaluation in figure5, the authors compared the untargeted vPro-MS with the highly sensitive targeted method - qPCR. For a fair comparison, is it possible to include sensitivity comparison results with the untargeted metagenomic method?

Authors' answer: We agree that, in an ideal world, comparing vPro with mNGS as the corresponding untargeted method would be the fairest comparison, and we have discussed this with our in-house mNGS experts. For mNGS the nucleic acids are often enriched or specifically depleted during sample preparation and the samples analyzed in our study were not enriched or depleted. Hence, for a fair comparison, one would need to analyze the samples in mNGS without enrichment. Unfortunately, there are still two major hurdles that we will not be able to overcome in a reasonable amount of time and effort:

i) Contamination is still a major problem in mNGS diagnostics, so a specific cut-off to distinguish a positive viral sample from a negative sample with minimal (cross-) contamination reads is mandatory. This is also one of the reasons why mNGS is not yet widely used for virus diagnostics. Therefore, Rollin and colleagues suggested in 2023 a novel tool for the identification of cross-contamination in mNGS called Cont-ID (<https://pubmed.ncbi.nlm.nih.gov/37833740/>). In support of the need for such a tool, they make the following statement: "As a single viral read could be detected in millions of sequencing reads, it is mandatory to fix a detection threshold that will be informed by estimated cross-contamination." However, tools like Cont-ID need a specific external control to determine the read thresholds and this not yet implemented in the mNGS in-house workflow.

ii) The patient samples used for analysis (SARS-CoV-2 panel) were collected during the pandemic and are therefore several years old. As SARS-CoV-2 is an RNA virus, the genome is rather unstable and it must be assumed, that the samples RNA has already been partially degraded, again making the comparison unfair. It must be noted here, that the qPCR results were generated at the time when the samples were collected.

For these reasons, it is not possible to make the suggested comparison between vPro and mNGS. But we have added the contaminations considerations to the discussion section and, moreover, confirmed the absence of cross-contamination in the vPro-MS workflow by running alternating measurements of positive and negative SARS-CoV-2 samples (see table S6 and text inserted in the discussion on page 22 (manuscript with tracked changes)).

7. In the effort to improve the sensitivity of vPro-MS, based on the author's description, the 75 synthetic peptide sequence were obtained from the vPro library and put through a further detectability filtering on samples, resulting in 65 peptides in the final library. (1) the synthetic library is still predicted by DIA-NN like the vPro library, how does the 65 peptides in the synthesized peptide library differ from those in the vPro library in terms of RT, IM and m/z? (2) Even though the synthetic library improved the LOD, it is insufficient to conclude that the improved sensitivity is attributed to synthetic library. Maybe it's due to the reduced library size? The authors could test this by taking the 75 SARS-CoV-2 specific

peptides plus the human proteome, creating a predicted library using DIA-NN, and reanalyzing the data to see if it also improves detection sensitivity. (3) Under the hood, DIA-NN performs target extraction of fragment ions based on the library, and the library quality heavily affects its performance, could using better prediction models instead of using DIA-NN improve sensitivity as well? (4) Even with the synthetic peptide library, the authors didn't mention whether all positive samples can be identified by vPro-MS.

Authors' answer: To address this point, we re-analyzed the SARS-CoV-2 sensitivity panel with 6 different libraries for this revision. The results are summarized in table 2. In order to analyze the influence of the library size on the sensitivity for virus identification, we restricted the initial vPro library (121, 977 viral peptides) to SARS-CoV-2 (177 viral peptides) or excluded all non-reference proteomes for HIV (53,600 viral peptides) during library generation. In another library, in conjunction with the SARS-CoV-2 restriction, human entries were restricted to peptides identified in the swab samples, which reduced the number of human entries compared to the reference from 591,159 to 18,655. This library size matches well with the size of the library obtained from the measurement of the synthetic SARS-CoV-2 peptides. In order to investigate the influence of the library prediction algorithm, we created another library from the same fasta files used for the initial vPro library prediction using the timsTOF model in AlphaPeptDeep. The results demonstrate, that the library size as well as the prediction algorithm have a minor effect on the sensitivity for virus identification. The library created from measurements of negative samples spiked with synthetic virus peptides clearly outperforms all in-silico predicted libraries. This library enables the detection of SARS-CoV-2 in 43 out of 58 PCR positive samples. Detailed numbers for all libraries are shown in table 2. Please note, that the library created from the measurements of samples spiked with synthetic peptides includes empirical values (RT, IM, type of fragments, fragment intensities), while the restricted databases contain predicted values. These new results are described in the manuscript on pages 19-20 (manuscript with tracked changes).

8. Is vPro-MS able to identify certain type or variants of virus? Taking SARS-CoV-2 panel for example, is it possible to identify the exact variant (alpha, delta or omicron)? The authors need to explicitly discuss this in the manuscript so that the readers will get aware of this limitation or advantage.

Authors' answer This is an important question to identify possible development directions of vPro. The sensitivity panel contained three different SARS-CoV-2 variants of concern (VOCs), namely alpha, delta and omicron. vPro-MS is currently not able to identify VOCs. First of all, the information needed to include VOCs in the vPro library is not available in UniProt. Nevertheless, we aimed to analyze if it could be possible. Therefore, we re-analyzed the data with the vPro library and additional entries of the three SARS-CoV-2 variants in our sample panel. The vPro results are available in the source data of table 2 and on PRIDE. The reports contain additional columns for the taxonomic layer "VOC". The VOC was correctly identified only in one delta sample by two unique peptides. If the peptide threshold is reduced to one, already 2 out of all 4 VOC identifications are wrong. The fact, that only in 4 out of 58 samples a VOC-unique peptide was identified demonstrates that this approach is not very promising. Maybe more sophisticated approaches might be suited for that purpose, but therefore a well curated protein sequence database of all VOCs is mandatory. The description of these results has been inserted on page 19 (manuscript with tracked changes).

9. As stated in page 15 line 409 'Protein sequences are more stable than viral genomes as mutations are only partially translated into gene products, which might be beneficial for virus detection in certain cases.' When a new variant occurs, immunoassay and qPCR may fail to identify the virus infection. Will vPro-MS have greater tolerance for identifying the virus infection regardless of the exact variant?

Authors' answer: We have added the following text on page 22 (manuscript with tracked changes): "In addition, untargeted proteomics is less prone to missed virus identifications as a result of emerging sequence variants. This is due to the nature of detection. While qPCR typically detects one or two target regions in the viral genome, untargeted proteomics is not restricted to the detection of relatively short target regions, but potentially covers all all tryptic peptides detectable by MS. It is also important to note that qPCR only works if the primer and probe binding sequences in the viral genome are intact. This means that they are not entirely mutated or deleted, which reduces the tolerance of qPCR to detect specific variants. An example is the new MPXV variant clade Ib, which has a deletion in the probe region of the qPCR assay from the CDC, which was used as reference assay prior to the occurrence of the new variant. Hence, to be able to detect the new variant, a new qPCR assay had to be established

⁴⁰. *The detection of the new MPXV clade Ib by vPro-MS would not have been affected by the deletion since numerous other peptides would have been available for identification.*”

10. Can the authors explain a bit why they decided to use a proteomes-wise level rather than the species or subspecies level when developing the vProID score for virus identification?

Authors' answer: This is an important question. During the development process of the vProID score different calculations were evaluated. However, the major problem was the large imbalance of the number of database peptides per species. 68,103 peptides map to HIV-1, while only 177 peptides map to SARS-CoV-2. As proteomic results always contain wrong identifications due to the target/decoy approach for evaluating search results, random HIV-1 identifications are much more likely than random SARS-CoV-2 identifications. Therefore, every score calculation has the problem, that it needs a signal above the presumed number of random hits. However, this severely decreases the sensitivity for viruses with many entries, because for HIV-1 many more peptides would be needed for identification than for SARS-CoV-2. One solution would be to limit the number of entries per species at a certain value keeping e.g. only the most conserved sequences. However, this balanced database does not reflect the sequence heterogeneity of the human virome anymore. When we calculate the vProID score per proteome, we completely circumvent this problem. As we only use structural proteins, the number of proteins per proteome is rather similar for different species. With this approach, we can keep analyzing the whole sequence space of the human virome without affecting the sensitivity.

Minor points:

1. In page 2 line 45-48, please provide references to the statements ‘One of the consequences is that data on the surveillance of viral infections is incomplete and that long-term effects of past infections are difficult to recognize’. As a reader of this manuscript rather than a virus expert, I am curious what kind of clinical cases suffered due to the incomplete surveillance of virus and how it will benefit from a comprehensive diagnostic?

Authors' answer: An example here is long COVID. At least 10 % of people develop long COVID symptoms after the infection with SARS-CoV-2. One third of long COVID patients do not have any identified pre-existing medical conditions. However, pathophysiology and risk factors are not fully understood (<https://www.nature.com/articles/s41579-022-00846-2>). Imagine having more complete surveillance data on people with long COVID, such as untargeted proteomics data from SARS-CoV-2 diagnostics. One could use the additional information beyond SARS-CoV-2 identification, e.g. the immune response, or analyse for co-infections or early markers in the development of long COVID. We have included this explanation in the Introduction so that the reader can follow our thinking behind this sentence (see page 2 in manuscript with tracked changes).

2. In page 2 line 71, please provide reference to ‘In microbiology, matrix-assisted laser desorption/ionization (MALDI) time-of-flight (ToF) mass spectrometry (MS) has become the method of choice for rapid, high-throughput and untargeted taxonomic classification of cultured bacteria.’ In recent years, many studies were conducted on the Orbitrap instruments. It’s hard to conclude which type is the method of choice. If the authors could refer to any supporting references that will help convince readers.

Authors' answer: There is no doubt about the method of choice in clinical microbiology. MALDI-ToF MS is used in thousands of clinical labs for routine testing, while none of them is using LC-MS-based proteomics for biotyping. This is obvious as the MALDI Biotyper from Bruker has an FDA clearance for diagnostics since more than 10 years (<https://ir.bruker.com/press-releases/press-release-details/2013/Bruker-Corporation-Announces-FDA-Clearance-to-Market-the-MALDI-Biotyper-CA-System/default.aspx>), while no LC-MS system has been approved by legal authorities for this purpose. Of course, there is a number of studies using LC-MS but this has no impact for routine diagnostics yet. We have included further references in the manuscript.

3. In page 3 line 89-91, it would be better to put it more straightforward by rephrasing the sentence in a way.

Authors' answer: The sentence has been rephrased.

4. For the virome spectral library construction, initially how many viruses in total were presented in the downloaded Uniport virome proteome database? Please address this in text or label it in figure2.

Authors' answer: The downloaded UniProt database contains 199 human-pathogenic virus species, of which 187 (94%) passed all filter steps and calculations and are included in the final vPro library. We included this information on page 9 (manuscript with tracked changes).

5. Please keep the library names consistent, 'vPro.Peptide.Library' was used in page 5 line 175, 'vPro.Viral.Peptide.Library' in page 5 line 193, while 'Viral.Peptide.library.txt' in page 8 line 266. Same for 'vPro.Virus.fasta' and 'VirusID.fasta'.

Authors' answer: The names have been changed to be consistent.

6. In figure4A, please explicitly label how many positive and negative samples in each dataset.

Authors' answer: This information is now shown in revised Fig 4A.

7. In figure4C, peptide number of MPXV is not labeled.

Authors' answer: We have removed the number of peptides for the labelled viruses, as this number is not used for validation of the virus identification. The revised figure 4 contains now only the vProID scores. The number of peptides can be found in the vPro results in the SI or on PRIDE.

8. In figure5, please color code the samples to distinguish positive from negative.

Authors' answer: This has been changed in the new version of Figure 5.

Reviewer #1 (Remarks on code availability):

The vPro-MS code is clean and easy to review. The implementation of the proposed score function aligns well with the described approach in the manuscript. A demo data is provided along with the code. Based on the provided readme file, no issues with installing and running the code. No results from the evaluation work are provided by the authors, which limits fully testing and reproducing the results in the manuscript.

Authors' answer: All Dia-NN reports, which were used as inputs for vPro are available on PRIDE with the dataset identifier PXD055135 (Username: reviewer_pxd055135@ebi.ac.uk and Password: 3XAHLqKFjzkwk). The vPro results are available on PRIDE and in the SI.

Responses to Reviewer #2

Reviewer #2 (Remarks to the Author):

The authors described in the manuscript entitled "vPro-MS enables identification of human-pathogenic viruses from patient samples by untargeted proteomics" the development of a strategy to perform untargeted identification of human viruses based on metaproteomics. This is subject of significant relevance and wide clinical applicability, not only on the surveillance of viral infections, but also, combined with classical virology methods such as cell culture and immunofluorescence, in daily routine of clinical laboratories. Its potential advantages and disadvantages compared to metagenomics based on next-generation sequencing are well explored and discussed throughout the text. To my knowledge, the application described in this manuscript is pioneer and its relevance is supported by the analysis of over 200 samples covering 18 different human-pathogenic virus species.

The objectives of the study are clearly identified, the methods are comprehensive in detail, the study presents its results and discussion with clarity and precision, and the conclusions are supported by the

data. It is a well written article, but there are some points that should be addressed before the manuscript can be accepted in this peer-review journal.

Authors' answer: The authors are grateful to reviewer #2 (R2) for taking the time to read the manuscript and are happy to implement the suggested improvements. In particular, we would like to thank R2 for the very important point he/she raised about the potential carry-over of the method. To address this point, we simulated an untargeted diagnostic routine workflow by alternating positive and negative virus samples. Using intermediate wash runs, we were able to demonstrate that no carry-over could be detected in the consecutive samples. This will be an important prerequisite for the use of untargeted proteomics in diagnostics.

“Introduction § 3rd: These studies rely on the use of data-dependent acquisition (DDA) on comparatively slow-scanning orbitrap instruments, which is currently being rapidly replaced for mere identification and quantification of proteins by DIA often performed on fast-scanning time-of flight (ToF) mass spectrometers.”

This whole sentence should be updated since newer orbitrap-based analyzers rival with ToFs in DIA in terms of speed, and the analyzer recently introduced with Astral is reported to be the faster for DIA analyses.

Authors' answer: We included the Astral in the sentence in the Introduction. The sentence now reads as follows: “These studies rely on the use of data-dependent acquisition (DDA) on comparatively slow-scanning orbitrap instruments, which is currently being rapidly replaced for mere identification and quantification of proteins by DIA performed using fast-scanning time-of flight (ToF) or Astral mass analyzers” (page 3 manuscript with tracked changes).

“Methods - Samples Preparation for Proteomics: Reduction and alkylation of cysteines was not included due to time constraints in this study. “

Explain what were the time constraints.

Comment the possible effects of including reduction and alkylation in the overall data quality and sensitivity of the strategy.

Authors' answer: We are aiming to make untargeted virus identification by proteomics as fast and convenient as possible. In our opinion this is the great strength of mass spectrometry compared to mNGS. It has been shown that the reduction and alkylation of cysteines does not affect the depth of proteomic analysis (<https://www.sciencedirect.com/science/article/pii/S0003267019314436>) and so this was the reason to skip this step. Therefore, skipping the reduction and alkylation of cysteines should not have an effect on the sensitivity of virus identification. Probably, this leads to slightly lower overall sequence coverage, but this is not the focus of this study. SARS-CoV-2 for example is identified in samples with low viral load solely by the nucleoprotein because this is the most abundant protein of this virus. The nucleoprotein however does not contain a single cysteine in its sequence (e.g. P0DTC9). The text on page 4 (manuscript with tracked changes) has been changed accordingly.

“Methods - vPro-MS Peptide Spectral Library: Afterwards, a peptide library of the remaining protein sequences was predicted using the deep-learning algorithm implemented in DIA-NN (version 1.8.1) with strict trypsin specificity (KR not P) allowing no missed cleavage site in the m/z range of 350 – 1,150 with charges states of 2 – 4 for all peptides consisting of 7-30 amino acids with enabled N-terminal methionine excision and cysteine carbamidomethylation. “

Why is cysteine carbamidomethylation informed to DIA-NN if this fixed modification was not included during sample prep?

Authors' answer: The publicly available data, which we downloaded from PRIDE, contains carbamidomethylated peptides. Furthermore, DIA-NN is trained on modified cysteines. In order to analyze all samples with a single library, carbamidomethylation was included in the analysis. It has also been shown that the reduction and alkylation of cysteines does not affect the depth of proteomic analysis (<https://www.sciencedirect.com/science/article/pii/S0003267019314436>).

“Afterwards, the following peptides were removed: (i) peptides, which were not specific to a certain virus species, (ii) peptides mapping to the human proteome, (iii) peptides mapping to common protein contaminants (BSA, trypsin, keratins). “

Provide more details in this section and include the scripts in the supplementary material.

Authors' answer: We have added the sentence "The human and contaminant peptide sequences used for filtering are available in the supplementary information along with the manually curated taxonomic information for each UniProt organism entry." to this section. The revised supplementary information contains a new folder called "vPro-MS Library Generation", which contains the aforementioned data as well as the R script used.

“Results - Assessing the confidence of virus identification: The number of specific peptides varied greatly between the different species within the library. HIV-1 was by far the largest part of the library with 68,103 peptides which equals 56 % of all entries. As peptides are usually identified in proteomics with a controlled error rate (often ~ 1 %), this means that few HIV-1 sequences were identified in almost every sample just by chance. “

Please discuss the reasons why HIV-1 is so overrepresented (e.g. number of proteins, variants, number of deposited sequences associated with clinical routine genotyping).

Authors' answer: In the current UniProt release 1,116,060 out of 1,465,021 proteins originating from human-pathogenic viruses are HIV-1 associated sequences (~75%). We don't know exactly why this large discrepancy exists and can only speculate about the reasons. We don't know the criteria of UniProt for including sequences in their database but suggest, that only formal criteria need to be fulfilled. We therefore believe, that this discrepancy results from the high mutation rate of retroviruses in general and the well-established genomic surveillance in developed countries, which in the end translates into a high number of entries in public sequence databases. We don't want that kind of speculation in the manuscript and hope this will be understandable to R2.

Could you comment on the effects of a hypothetical HIV data curation/reduction (e.g. selecting the most detectable HIV peptides) to balance the library?

Authors' answer: We have created a library, which includes only the reference HIV proteomes in UniProt. This reduces the number of virus peptides in the vPro library from 121,977 to 53,600. However, this reduction has only a minor effect on the sensitivity of the workflow. The results are summarized in the new table 2 and the underlying data can be accessed in the source data.

“Evaluation of the sensitivity of vPro-MS for the detection of SARS-CoV-2: Virus quantification by proteomics and qPCR correlated with an R^2 value of 0.62.”

Please compare the correlation (r^2) of vPro-MS and qPCR for the detection of SARS-CoV-2 with previous described targeted assays (which also included correlation with qPCR).

Authors' answer: Analogous comparisons (protein intensity vs. Ct value) have been done, for example, in PRM studies targeting SARS-CoV-2:

- *Hober A et al. (2021) Rapid and sensitive detection of SARS-CoV-2 infection using quantitative peptide enrichment LC-MS analysis. eLife*
- *Lane D et al. (2024) A high throughput immuno-affinity mass spectrometry method for detection and quantitation of SARS-CoV-2 nucleoprotein in human saliva and its comparison with RT-PCR, RT-LAMP, and lateral flow rapid antigen test. Clinical Chemistry and Laboratory Medicine (CCLM)*
- *Cardozo, K.H.M. et al. (2020) Establishing a mass spectrometry-based system for rapid detection of SARS-CoV-2 in large clinical sample cohorts. Nat Commun*
- *Renuse S et al. (2021) A mass spectrometry-based targeted assay for detection of SARS-CoV-2 antigen from clinical specimens. eBioMedicine*
- *Mangalaparthy, K.K. et al. (2021) A SISCAPA-based approach for detection of SARS-CoV-2 viral antigens from clinical samples. Clin Proteom*

In these studies, the correlation (R^2) between viral protein amounts and Ct values ranged from 0.54 to 0.82, depending also on the target peptide and potential enrichment techniques applied during sample preparation. A correlation of 0.62, as shown in our study, can therefore be considered to be within the expected range. This information was added to the manuscript on page 19 (see manuscript with tracked changes).

“Discussion: However, it is essential to wash the LC columns between samples. Samples are measured and analyzed in a consecutive order.”

It is crucial to provide experimental evidence (supplementary section) demonstrating that a single blank injection between samples effectively eliminates carry-over effects. To validate this, we recommend reinjecting low Cts SARS-CoV-2 followed by one, two, or three negative controls, and subsequently analyzing the raw data through the established processing workflow.

Authors' answer: This is a very important question and we aware, that carry-over effects would be devastating for this diagnostic approach. We were dealing with this issue for many years but the LC-MS setup used in this study is able to overcome the problem. In order to prove this, we have conducted an experiment with alternating measurements of SARS-CoV-2 positive and negative swab samples to assess potential carry-over effects. The viral loads of the positive samples were varied. In between each sample the Evosep wash method was used to flush the system. As expected, no carry-over effect was detected and no virus was identified in the negative samples. The vPro results table is presented as table S6 and the experiment is mentioned on page 22 (manuscript with tracked changes).

Responses to Reviewer #3

Reviewer #3 (Remarks to the Author):

The manuscript by M. Grossegeisse et al. entitled “vPro-MS enables identification of human-pathogenic viruses from patient samples by untargeted proteomics”, describes an interesting approach to mine DIA-based L/MS data to identify viral proteins to aid the diagnosis of viral infections, and to quantify the viral titer (or a proxy thereof). The chosen approach is to analyze purified virus samples, and to curate the known viral protein sequences to come up with a vetted spectral library with peptides from capsid, envelope, virus membrane proteins, supplemented with real data from a subset of viruses.

While the problem addressed in this manuscript is highly relevant and timely, and the chosen approach is interesting and promising, the manuscript is not yet in a shape that would allow for an accept/reject decision.

Authors' answer: The authors are grateful to reviewer #3 (R3) for the time and care taken in reviewing the manuscript. The authors acknowledge that the potential value of the manuscript was recognized by R3 and that the questions raised are likely to be the same questions that future readers would ask. We have therefore undertaken an intensive revision of the entire manuscript, including harmonizing numbers and figures and improving explanations in the text.

As a short comment to the summary text of R3, we firstly would like to emphasize that the study is based on the analysis of patient samples and not purified viruses, which is already highlighted in the title of the manuscript (“vPro-MS enables identification of human-pathogenic viruses from patient samples by untargeted proteomics”). This makes a huge difference for evaluation of the potential impact of this manuscript. We hope that we have made this point clearer in the revised manuscript. Secondly, we also see the potential to quantify or at least approximate viral titers in patient samples using DIA-MS. As demonstrated by the top3 approach in our study, it is possible to correlate the amount of the three most abundant peptide species of a virus to the Ct value. However, the quantification in patient samples highly depends on the sample material itself and the manner of sampling. As an example, take the different quantitative nature of swab samples and serum samples. There is no amount of sample material that can be used to quantify a swab sample, whereas serum can be collected in clearly measurable quantities (e.g. mL). Moreover, a swab sample may be taken precautionary or more rigorously, resulting in different amounts of sample material and therefore potentially different virus amounts. In PCR diagnostics, the sample material may be approximated by parallel analyzation of housekeeping genes (e.g. c-myc). In proteomics-based diagnostics one may use the human protein amount in the sample to approximate the amount of material in the sample. The advantage of DIA-MS (and therefore the vPro-MS workflow) is that the complete human background is analyzed in parallel. Although there are

currently no marker proteins established, by using vPro in a larger sample cohort, one could correlate different housekeeping proteins found in the sample to qPCR marker genes to establish a reference for sample amount quantification or as internal marker for sample quality. While this is beyond the scope of this manuscript and may be the subject of further research, it underlines the potential of DIA-based virus detection from patient samples. After careful review of the manuscript, the authors believe that the changes suggested by R3, particularly the clarification of numbers and the calculations behind them, have significantly improved the manuscript and hope that this will facilitate the decision process.

The following issues (in a somewhat arbitrary order) have to be addressed before decisions can be considered:

1) The manuscript is at times hard to follow also due to numerous inconsistencies in the numbers.

Authors' answer: We are sorry that R3 sees it this way. After re-reading the text and taking the time to read it through, we agree with R3 in parts. As a result, the manuscript has been extensively revised with a focus on clarity and comprehensibility to enable the reader to follow the authors' reasoning and calculations. We believe that this extensive revision significantly improved the manuscript.

a. For example, the authors keep stating 331 virus species, but Table S1 lists only 187.

Authors' answer: The 331 viruses refer to different viruses retrieved from UniProt as a result of lowest common ancestor analysis after applying different protein and peptide filters (see Fig.2). The 331 viruses include 187 viruses which can be distinguished on species-level. As the different levels of viral taxonomy are not well annotated in the UniProt database, all further taxonomic information apart from the species (e.g. the clade or serotype of a virus) was aggregated into a subspecies rank. This subspecies is not an official virus taxonomy categorization as it simply summarizes all taxonomic information below the species rank. Nevertheless, it seemed to us to be the most application-oriented approach to preserve the database information for reporting. To make this clear, this resulted in a species-level annotation for viruses that do not necessarily have a subspecies information. The other way around, all subspecies information is connected with a species information. This is why it is also possible that there are more different subspecies (193) than species (187) in the vPro peptide library. If both levels are combined, a total of 331 viruses can be distinguished with vPro-MS (see table S7). An example is the identification of SARS-CoV-2. SARS-CoV-2 is not a virus species. It is a subspecies of the species "Severe acute respiratory syndrome coronavirus" in UniProt, which includes also the subspecies SARS-CoV, which caused the outbreak in Asia in 2002. Please note, that there is no such virus as SARS-CoV-1. For historical reasons species and subspecies "SARS-CoV" have exactly the same name which is quite misleading but is consistent with the database of the International Committee on Taxonomy of Viruses (<https://ictv.global/>). Our data show, that vPro-MS is able to distinguish subspecies SARS-CoV-2 from subspecies SARS-CoV in all analyzed samples and correctly reports SARS-CoV-2 in the subspecies column, while the species column states SARS-CoV, which is also correct.

b. They list all the viruses they have tested in the material and methods section; unless the authors really want the reader to count the listed viruses, it might be a good idea to list the viruses tested in a supplementary table.

Authors' answer: We have shortened the virus list in the material and methods section and added a supplementary tables (table S1 and S2) containing all viruses used in this study.

c. Statements like "When applying a 1 % FDR on precursor-level, 959 viruses were incorrectly identified, which equals ~ 14 viruses per sample" also don't help if only 331 viruses are covered. I can make assumptions about the origin of these 959, but the authors should not rely on assumptions from the reader, but should clearly describe what they mean.

Authors' answer: This is a valid point. One major difference between this study and most proteomics studies is that for virus diagnostics each sample needs to be treated completely independent from each other. This means, that each sample is tested for 331 viruses independently. Therefore, the same virus

can be incorrectly identified in multiple samples and is therefore counted multiple times, because each identification in itself is wrong. The sentence was deleted from the manuscript as the whole section was re-written see page 10-11 (manuscript with tracked changes).

d. For Figure 4, the authors state “The specificity of the vPro-MS workflow was determined by the analysis of 221 samples from 4 different sources covering 18 human-pathogenic viruses (Fig. 4).” Looking at panel 4c, it was not clear what T1 to T19 stood for. Never mind N20 to N27. Eventually, it dawned on me, that this is the specificity panel, which has 19 positive samples and 8 negative control samples, which was not clearly explained. In short, a much better explanation of the different datasets used for Figure 4 is needed.

Authors’ answer: The authors agree that the reader needs more information in order to understand the data on which Figure 4 is based. Therefore figure 4 has been revised and now includes more information on the sample panels, e.g. sample numbers, sample types, MS analysis details and viruses included. We have further added tables S1 and S2, which describe the sample panels measured in this study with more details. In the specificity panel “T samples” are positive for viruses (“target”) and “N samples” are negative.

e. Often, the authors don’t provide all the numbers and/or clear instructions which values were used to calculate certain outcomes. Case in point A) is Table S2: please provide all the values and clear instructions, so that I can recalculate the vProID score. Currently, I have to gather information from various pages. Similarly, Figure 3: plenty of numbers are given, but based on the description, I am not completely clear which numbers made it into the calculation of specificity. Again, I can make assumptions, but that is always a bad starting point, clearly showing that the manuscript is not good enough in its current state.

Authors’ answer: Table S4 is an intermediate result of the vPro result for sample T17, which includes additional information extracted from the vPro library to show how vPro actually works. Unfortunately, it is not possible to publish all of the intermediate results in the manuscript. In order to underline the performance of the vProID score we have extensively revised figure 3 and included a new table 1. We conducted additional data analysis to compare various filtering strategies for the analysis of the SARS-CoV-2 respiratory swab sample panel. In total, 16 different parameter sets were tested either with or without using vProID scores (32 results in total). In case you still would like to recalculate the Table S2 (now S4) you need to use the script provided on Github (<https://github.com/RKI-ZBS/vPro-MS>) and analyze the DIA-NN report of the specificity panel, which is provided at PRIDE (see data availability statement). You would then need to stop the script before the proteome information is summarized for each species. The information on the number of theoretical peptides per proteome or species can be obtained from the vPro.Peptide.Library.txt, which is either available at PRIDE (see data availability statement) or at Zenodo (<https://zenodo.org/records/13832021>). The Zenodo link is shared at the github repository. The underlying data of figure 3 are now available in the SI source data figure 3).

Speaking of Figure 3: this seems to be a bit of a weak spot. Just comparing the specificity and sensitivity for 0.1% FDR vs. vPro, the values are 99.8% vs. 100% and 34/58 vs. 32/58, i.e., these numbers are so similar that I would not dare to say that vPro is so much better than just tightening the FDR. If these numbers are the best that the authors can muster, I am not sure whether this approach, despite being interesting, warrants publication in Nature Communications.

Authors’ answer: In order to better highlight the improvements made possible by vPro-MS, we conducted additional data analysis to compare various filtering strategies for the analysis of the SARS-CoV-2 respiratory swab sample panel. In total, 16 different parameter sets were tested either with or without using vProID scores (32 results in total). We present the results in the completely revised figure. 3 and the new table 1. In that way it becomes clearer, that regardless of the other parameters, applying the vProID score significantly improves specificity, reduces false positive virus identifications and enhances sensitivity. Please note, that the results differ from the ones presented in the original figure 3. In the first version of the manuscript, the various filter parameters were applied to the data via the vPro-MS analysis script. However, this meant, that the minimum number of peptides per virus was calculated per proteome (UniProtID) and not per species/subspecies. The revised manuscript now contains results with vPro-MS completely turned off, as one would be able to generate from the DIA-NN output by obvious filtering strategies such as min. number of peptides per virus species. This leads to an increase

of unspecific virus identifications compared to the original manuscript. The virus identification results for all parameter sets can be accessed via the source data folder in the SI. An overview of the peptide identification results separately listed for human, virus and contaminant sequences for each parameter set is given in Table 1. This summarizes the data of 2112 (32 parameters x 66 samples) single sample results.

The authors introduce some concepts, which are not necessarily easily understood for the majority of the proteomics specialists, which handle mammalian samples, which normally assume that one organism has one proteome. This concept does not apply to viruses and thus need some explanation so that everybody can follow.

Authors' answer: Viruses are genetically highly diverse and by adaptation they continually change. Viruses are grouped by their general characteristics, like genome organization and particle form, but mainly by their genome sequence. The International Committee on Taxonomy of Viruses (ICTV) classifies viruses into different hierarchical levels of order, family, subfamily, genus and species. Species, in turn, can contain different stable genetic variants called strains or clades. More transient changes in the genome may be simply called a variant. It is not unusual that viruses become re-classified or additional subspecies or strains are added. We recognize that this wealth of taxonomic information can be difficult for non-virologists to understand. In our study we use the species annotation of the UniProt database as the highest taxonomic order and group all other taxonomic information in a subspecies group. Virus species in the vPro library contain a single or even up to several thousand associated proteomes, like in the case of HIV. This difference in the number of proteomes per species is partly due to a certain heterogeneity in the species itself, but also to the attention paid to a particular virus and the amount of research effort that goes into it. We have added an explanation of the virus taxonomy to the part "Construction of the human virome peptide spectral library" to make the concept more accessible to non-virologists (see page 9 in manuscript with tracked changes).

Similarly, the 'virus level' and 'sample level' specificity used in Figure 4 is not well described. I thought I had understood it, but the fact that the plasma study (PRIDE) listed 'no data' for the sample level specificity clearly drove home the message that I did NOT understand the difference. Again, better/clearer explanations are clearly needed.

Authors' answer: Specificity is defined as the ability of a test to identify true negatives. As samples were tested for multiple targets (331 viruses) specificity of vPro-MS was calculated either referring to samples or viruses. Specificity was defined as the number of true negative (TN) virus tests or samples divided by the sum of true negative and false positive (FP) virus tests or samples (specificity = $TN/(TN+FP)$). All numbers are listed in revised figure 4A. The plasma study (PRIDE) does not contain negative samples, so we are not able to calculate specificity on sample-level.

In the vProID Score formula, it is not clear whether "#peptides(sample)" refers to peptides, unique peptides, PSMs or MS2 spectra. Please be more specific.

Authors' answer: As stated in the manuscript, the vProID score is the log10 value of the number of identified peptides for a given virus proteome in a certain sample divided by the expected number of peptide identifications for this virus proteome just by chance in this specific sample. The vPro score is calculated on peptides identified using the vPro peptide spectral library, in which peptides that are not specific to a certain virus species are removed as well as peptides matching the human proteome or common contaminants. Hence, the score is based on unique peptides.

The vProID score is introduced and explained in great detail (though not very comprehensible at times). However, in Figure 3 and 4, the meat of the paper, the vPro ID score is not used any longer. Some explanation for this decision to omit further use of the vProID score might be useful.

Authors' answer: The authors understand that it may appear at first glance that the vPro score is not used, but no, this is not the case. In figure 3 the vProID score is introduced. We have completely revised figure 3 and added a visualization of the vPro score distribution to the revised figure 3B. We further guess that R3 is referring to figures 4 and 5 as "the meat of the paper". These figures show the results of sensitivity and specificity evaluation of vPro-MS. Of course, the vProID score was used in all cases. Although not explicitly shown, the vPro score is the basis for all results shown. In the methods section

under the heading “vPro-MS – Virus Identification” we wrote the sentence “The threshold for virus identification used in this study was set to a minimum of 2 peptides and a vProID score of 2.” This means, that in this study a minimum vProID score of 2 was required for virus identification in each sample. We have added this explanation of the minimum vPro score for virus identification to the figure legend of Figure 4 to make it easier for readers to understand. The actual scores for each sample are available in the vPro reports, which are accessible as supplementary information as well as via the PRIDE repository.

Figure 5: the authors show some decent correlation between protein intensity (log transformed) and Ct value. Assuming that the protein intensities are Log 2 transformed (which is not clearly stated as y-axis label), it is not clear as to why the slope is not 1. Can the authors speculate/discuss this, as the Ct value should also be the equivalent of a log2 transformation, i.e., the linear fit should have a slope of 1?

Authors’ answer: We added the log2 transformation information to the y-axis. It is important to remember that we are comparing completely different methods that detect completely different targets (viral RNA versus viral protein). Due to the different nature of the targets, sample preparation and analysis methods one would not expect a correlation of 1 in such a comparison. Analogous comparisons (protein intensity vs. Ct value) have been done e.g. in PRM studies targeting SARS-CoV-2:

- Hober A et al. (2021) Rapid and sensitive detection of SARS-CoV-2 infection using quantitative peptide enrichment LC-MS analysis. *eLife*
- Lane D et al. (2024) A high throughput immuno-affinity mass spectrometry method for detection and quantitation of SARS-CoV-2 nucleoprotein in human saliva and its comparison with RT-PCR, RT-LAMP, and lateral flow rapid antigen test. *Clinical Chemistry and Laboratory Medicine (CCLM)*
- Cardozo, K.H.M. et al. (2020) Establishing a mass spectrometry-based system for rapid detection of SARS-CoV-2 in large clinical sample cohorts. *Nat Commun*
- Renuse S et al. (2021) A mass spectrometry-based targeted assay for detection of SARS-CoV-2 antigen from clinical specimens. *eBioMedicine*
- Mangalaparthy, K.K. et al. (2021) A SISCAPA-based approach for detection of SARS-CoV-2 viral antigens from clinical samples. *Clin Proteom*

In these studies, the correlation (R^2) between viral protein amounts and Ct values ranged from 0.54 to 0.82, depending also on the target peptide and potential enrichment techniques applied during sample preparation. A correlation of 0.62, as shown in our study, can therefore be considered to be within the expected range. This information was added to the manuscript on page 19 (see manuscript with tracked changes). Please also note, that this correlation is also similar to the results reported in studies comparing mNGS and qPCR (see table S5).

In summary, this manuscript has potential, but needs major revisions. The current version is hard to follow, and not well structured. The authors might want to rethink which experiments are truly needed to drive their main message home. Similarly, they should enable the readers to follow their arguments and their calculations without the need of gathering information on various pages and/or having to make assumptions.

Authors’ answer: The authors would like to thank R3 for taking a critical look at the manuscript. This has helped to improve the manuscript and hopefully make it easier to follow and understand.

Point-by-point response to the reviewers' comments

Dear Reviewers,

Thank you for the time and effort you have dedicated to reviewing our manuscript. We have carefully revised the figures, addressing all comments and suggestions. We have clarified points on GitHub where there were questions and adjusted accordingly.

Thank you again for your time.

Sincerely,

Dr. Marica Grossegeesse
Dr. Fabian Horn
Dr. Andreas Kurth
Dr. Peter Lasch
Prof. Dr. Andreas Nitsche
Dr. Joerg Doellinger

Responses to Reviewer #1

REVIEWER COMMENTS

Reviewer #1 (Remarks to the Author):

The authors have addressed all my comments with expanded evaluations and detailed responses. I have a few minor suggestions that may help improve the visualization of their figures, particularly Figure 3, 4 and 5.

In Figure 3, 1) consider making the axes black instead of gray; 2) consider adding parameter combination information at the bottom of the x-axis in Figure 3C, which can be organized into three rows, each representing one parameter type. This change would help clarify what each parameter set represents, making it easier to interpret the results; 3) please consider adding corresponding dots to the line chart so that the position of each data point is clearer.

In Figure 4C, please modify the color scale to make it more distinguishable across different score levels. In Figure 5, if there is no specific reason for using the x-axis limits of 15 to 40, please consider making plots more compact to reduce empty space.

Other than that, I have no further concerns.

Authors' answer: All figures have been changed according to the suggestions.

Reviewer #2 (Remarks to the Author):

I appreciate the authors' efforts in carefully addressing all the comments and suggestions provided in the first round of review. The authors have provided well-justified responses to each reviewer comment, with appropriate modifications in the text. The revisions have significantly improved the clarity, rigor, and overall quality of the manuscript.

The revised Table 1 notably enhances the manuscript by clearly illustrating the impact of filtering parameters on virus identification, providing much-needed transparency in this analytical approach.

Minor Corrections Suggested:

During my evaluation, I tested the scripts using both the provided demo data and other DIA-NN data freshly generated in my lab and I identified minor inconsistencies that should be addressed in the final version to ensure vPro-MS reproducibility.

- 1) Line 4 (`use_mit_license()`) had to be commented out for successful installation/execution in R, as it threw an error in a standard R environment.

Authors' answer: This issue has been resolved.

- 2) The default DIA-NN main report was not in a directly usable format for the pipeline. I needed to reformat my lab's DIA-NN-generated TSV files to match the structure of vPro-MS/demo/data.txt. Similarly, DIA-NN predicted library outputs required restructuring to align with demo/library.txt format.

Authors' answer: We apologise for the inconvenience. This was a misunderstanding: vPro can handle DIA-NN-generated TSV files directly. However, the demo data has a different structure because all the unused columns have been removed from the DIA-NN main report. This is necessary because the file size limit on GitHub is very low. vPro will only load the columns specified in the demo data and ignore all others. Therefore, it is not necessary to remove them for the analysis. Furthermore, the demo/library.txt file is not a DIA-NN-generated library, but rather the vPro.Peptide.Library.txt file. The latter is available via Zenodo (see the link on GitHub). It is available on Zenodo because of GitHub's file size limitations. To further clarify this issue, we have renamed the demo files and expanded the accompanying text on GitHub.

- 3) While the provided Source material (546235_1_data_set_10646140_smr62m\Source data\vPro-MS Library Generation\Library Calculation.R) gives some clues about using tidyverse for this manipulation, these preprocessing steps should be integrated into vPro.R (as expected based on the manuscript) or provided a standalone preprocessing script.

Authors' answer: Library construction (Library Calculation.R) is not really a pre-processing step and cannot be integrated into vPro.R, because the correct running order is:

Library construction (Library Calculation.R) -> Peptide identification (DiaNN) -> Virus identification (vPro.R).

Library construction (step 1) cannot be integrated into virus identification (step 3) because DiaNN needs to be run in between. Furthermore, the library only needs to be constructed once. The final library is included in the publication and is available via Zenodo (see the link on GitHub). There is no need to construct a new library for each sample. The same vPro library can be used for every analysis, and its performance (specificity and sensitivity) is validated in the publication.

Reviewer #2 (Remarks on code availability):

Details are provided in the comments above for the authors.